# Rethinking Verification for LLM Code Generation: From Generation to Testing

**Zihan Ma**[1,2,3,*]**, Taolin Zhang**[1,*]**, Maosong Cao**[1]**, Junnan Liu**[1]**, Wenwei Zhang**[1]**,
**Minnan Luo**[2,3,†]**, Songyang Zhang**[1,†,‡]**, Kai Chen**[1,†]

[1]Shanghai AI Laboratory
[2]School of Computer Science and Technology, Xi'an Jiaotong University, China
[3]MOE KLINNS Lab, Xi'an Jiaotong University, China
`{mazihan880}@stu.xjtu.edu.cn`
`{zhangtaolin,zhangsongyang}@pjlab.org.cn`
`{minnluo}@xjtu.edu.cn`

## Abstract

Large language models (LLMs) have recently achieved notable success in code-generation benchmarks such as HumanEval and LiveCodeBench. However, a detailed examination reveals that these evaluation suites often comprise only a limited number of homogeneous test cases, resulting in subtle faults going undetected. This not only artificially inflates measured performance but also compromises accurate reward estimation in reinforcement learning frameworks utilizing verifiable rewards (RLVR). To address these critical shortcomings, we systematically investigate the test-case generation (TCG) task by proposing multi-dimensional metrics designed to rigorously quantify test-suite thoroughness. Furthermore, we introduce a human-LLM collaborative method (SAGA), leveraging human programming expertise with LLM reasoning capability, aimed at significantly enhancing both the coverage and the quality of generated test cases. In addition, we develop a TCGBench to facilitate the study of the TCG task. Experiments show that SAGA achieves a detection rate of 90.62% and a verifier accuracy of 32.58% on TCG-Bench. The Verifier Accuracy (Verifier Acc) of the code generation evaluation benchmark synthesized by SAGA is 10.78% higher than that of LiveCodeBench-v6. These results demonstrate the effectiveness of our proposed method. We hope this work contributes to building a scalable foundation for reliable LLM code evaluation, further advancing RLVR in code generation, and paving the way for automated adversarial test synthesis and adaptive benchmark integration. [1]

## 1 Introduction

Large Language Models (LLMs) have triggered a paradigm shift in automatic code generation, demonstrating capabilities on par with or even exceeding human programmers on numerous benchmark tasks. As LLMs become increasingly integrated into software development workflows, ensuring the quality and reliability of the code they produce is paramount. This necessitates reliable evaluation methodologies, where code verifiers—typically powered by test suites—play a critical role. This raises a crucial question: *Are the test cases of current benchmarks for evaluating models' code capabilities robust enough?*

Initial analyses of benchmarks like HumanEval [6] (avg. 7.7 tests/problem), MBPP [3] (3 tests/problem), and EvalPlus [28, 29] (which saw a 15% pass rate drop with 80× more tests) indicate the

---

† means corresponding authors, ‡ means project lead, ∗ means authors contributed equally.
[1]The data demo and prompts can be accessed via https://github.com/open-compass/SAGA

fragility of current evaluation setups due to sparse test coverage. While methods like TestEval [43] tailor tests to specific solutions, they are inefficient for large-scale evaluation and impractical for dynamic integration into RL training loops, thereby hindering the development of models robust against diverse failures. LiveCodeBench [20] employs LLMs to generate numerous tests from golden solutions and synthetic inputs, aiming to enhance robustness.

However, a fundamental concern is that such methods may inadvertently create tests biased towards typical, often homogenized, LLM error patterns, which starkly contrast with diverse human reasoning errors. Conversely, competitive programming platforms (Online Judges) possess extensive, rigorously curated test suites for assessing code robustness, but these are often private and inaccessible. This underscores the urgent need for accessible and robust code verifiers to enable reliable performance evaluation and reward estimation.

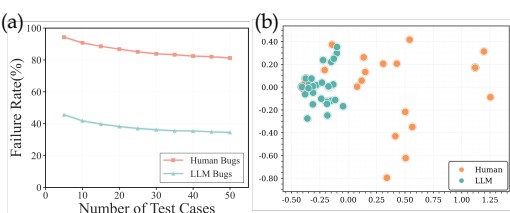

Figure 1: (a) Verifiers synthesized primarily from LLM-generated data exhibit a high failure rate when testing human-written bugs. (b) PCA analysis reveals that LLM-induced errors are highly clustered, indicating systematic weaknesses, whereas human errors are diverse and dispersed, posing greater challenges to existing verifiers.

In this work, we first identify limitations in current benchmarks through preliminary experiments. Our investigation reveals critical weaknesses in existing verifier suites. Specifically, when we took LLM-generated solutions that had passed LiveCodeBench's private tests and re-evaluated them on LeetCode's online judge, we found that for a significant portion of these solutions—20% for medium and 40% for hard problems, respectively—LeetCode identified errors that LiveCodeBench's verifier had missed. This demonstrates that LiveCodeBench's verifiers can be flawed, failing to achieve comprehensive error detection and thereby overestimating the true quality of the LLM-generated solutions. Furthermore, these LLM-centric verifiers themselves exhibit a high failure rate (i.e., they fail to detect existing bugs) when evaluating human-written faulty code, a rate much higher than any apparent failure rate on LLM-generated errors (Figure 1(a)). PCA analysis of error patterns (Figure 1(b)) reveals that LLM errors cluster tightly, indicating shared systematic biases, while human errors are widely distributed across a complex error landscape.

These findings underscore the fundamental inadequacy of constructing verifiers solely from LLM-generated data, as such approaches misassess LLMs' coding capabilities and provide flawed training feedback, manifesting two critical challenges: **(1) Test Case Homogenization and LLM-Centric Bias**: LLM-based TCG methods produce test suites that mirror the generating models' error patterns and cognitive biases, creating a "homogenization trap" where tests focus on LLM-like failures while neglecting diverse human programming errors (e.g., logical flaws, integer overflows). **(2) Verifier Ineffectiveness and Persistent Blind Spots**: Verifiers built on such test suites exhibit blind spots for human-like errors, failing to rigorously evaluate code due to challenges in generating tests for complex boundary conditions and interaction scenarios, compounded by LLM-centric test design. These challenges impede reinforcement learning frameworks (e.g., DeepSeek-R1 [8], O1-IOI [10]) from leveraging verifiable rewards, leading to *optimization misdirection* via *reward hacking*[2].

To address these challenges, we develop a comprehensive measurement framework for code verifiers, introducing multi-dimensional evaluation metrics (detection rate, verifier accuracy) and identifying test case diversity and per-case strength as critical quality factors. We derive an upper bound for detection rate and validate it empirically across 1,500+ coding problems. Using this framework, we uncover systemic test quality issues in leading benchmarks, such as CodeForce-CoTs [39], where 50% of problems had tests failing to detect known errors and 84% of verifiers were flawed, further underscoring these limitations.

To systematically improve the quality of code verifiers, we formally define the Test Case Generation (TCG) task, comprising three core components: *task formulation*, *benchmark construction*, and *method exploration*. We introduce TCGBench, a benchmark curated by aggregating representative problems from three leading competitive programming platforms—Atcoder, Codeforces, and Nowcoder. TCGBench curates human-verified adversarial examples spanning diverse error patterns (e.g.,

---

[2]Models exploit verifier weaknesses instead of achieving genuine correctness, as they are insufficiently penalized for diverse errors.

logical flaws, edge cases) and supports comprehensive evaluation of TCG methods. We further investigate current TCG methods and propose SAGA (**S**trategic **A**dversarial & Constraint-differential **G**ener**A**tive workflow), a novel human-LLM collaborative framework. SAGA is designed to systematically generate high-coverage, highly discriminative test cases by leveraging both human-derived constraints from correct solutions and insights from failure modes in incorrect solutions. This dual-pronged analytical approach allows SAGA to achieve improvements over current state-of-the-art TCG methods, with the Detection Rate increasing by 9.55% and Verifier Accuracy by 12.14%.

Additionally, we leverage SAGA to enhance the quality of the popular code generation benchmark LiveCodeBench-v6 (subset)[3], and develop CodeCompass, a new high-quality code benchmark. We believe SAGA can further be employed to scale datasets, enabling the production of training data with robust and accurate reward estimation for coding tasks. Our main contributions are as follows:

- We construct **TCGBench**, a comprehensive dataset from competitive programming platform, to analyze existing Test Case Generation practices. On it, we formalize multi-dimensional metrics for rigorous test case quality evaluation.
- We propose and validate **SAGA**, a novel human-LLM collaborative TCG framework (Fig. 5). By integrating insights from both correct and incorrect human solutions, SAGA generates significantly more effective test suites, improving Verifier Accuracy by 15.86% over existing TCG methods.
- Using SAGA, we develop **CodeComPass**, a challenging benchmark with human-verified adversarial examples for robust code generation evaluation, and introduce **TCGCoder-7B**, a SAGA-distilled specialist model for capable TCG.

## 2 Related Work

Reliable evaluation is paramount for advanced code-generating LLMs [37, 13], especially in Reinforcement Learning from Verifiable Rewards (RLVR) contexts [8, 31]. However, existing benchmarks often exhibit limitations such as LLM-centric biases and insufficient test diversity [28, 20]. This underscores the need for more effective **Test Case Generation (TCG)**. Current LLM-based TCG methods, primarily *Direct Generation* [23] and *Input-Interpreter* [20], still struggle to produce comprehensive and challenging test suites that cover subtle corner cases. Our work, SAGA, addresses this gap by introducing a novel human-LLM collaborative framework designed to systematically generate higher-quality tests by leveraging deep human insights. A comprehensive review of related works is provided in Appendix A.

## 3 Evaluating Verifier Quality: Metrics and TCG Paradigms

The reliable evaluation of LLM-generated code is critically constrained by the quality and accessibility of verifiers. Standard benchmarks like HumanEval [6] often employ limited test suites, while the extensive private verifiers of Online Judges remain largely inaccessible for broader research. This scarcity of robust, accessible verifiers impedes accurate LLM evaluation and the advancement of RLVR. To surmount this challenge, we focus on leveraging LLMs themselves for **Test Case Generation**—the systematic synthesis of test suites. Figure 2 illustrates the pivotal role of a Code Verifierin the LLM code evaluation pipeline. It also highlights distinct TCG paradigms:

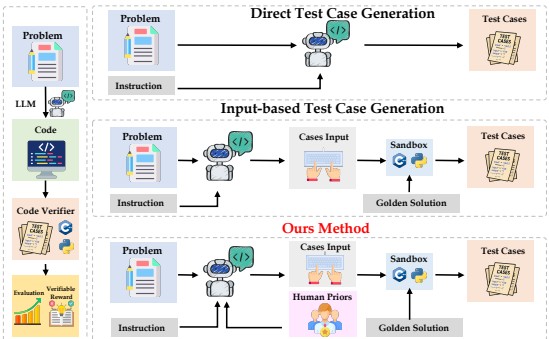

Figure 2: The code evaluation pipeline and different TCG paradigms.

---

[3]The subset used in our study comprises 101 problems from AtCoder, specifically from contests ABC387 through ABC400 and ARC190 through ARC196 (problems A to F). This selection facilitates a consistent evaluation scope when comparing with benchmarks like our proposed TCGBench, which also incorporates these recent algorithmic challenges.

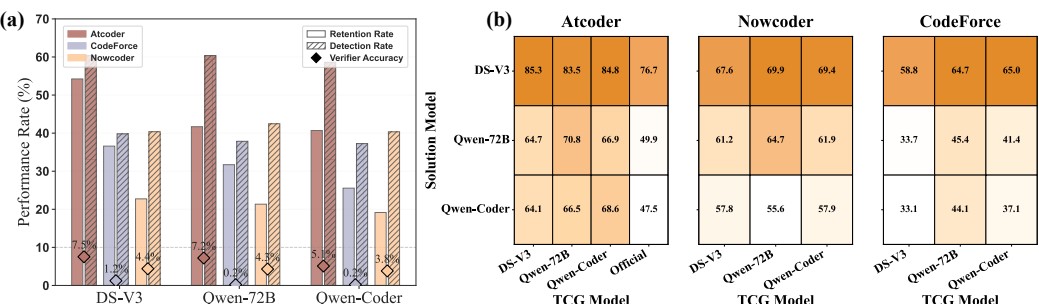

Figure 3: Direct generation issues: (a) Low quality of LLM-generated tests. (b) High self-pass rates suggest model blind spots.

- **Direct Generation:** An LLM directly produces complete test cases (inputs and outputs). Representative works include TestChain [23], AceCoder [48], and CodeRM [33].
- **Input-Interpreter:** An LLM generates test inputs; a ground-truth interpreter (or reference solution) then computes the corresponding outputs. This paradigm, exemplified by LiveCodeBench [20] and Codeforce-COT [39], often employs random input sampling—a strategy proven effective by Live-CodeBench for generating numerous diverse test cases and thus adopted in our baseline evaluations. EvalPlus [28], which mutates seed inputs for execution, also aligns with this approach. A comparative analysis with EvalPlus is detailed in Section 4.2.1.
- **Human Priors (Our Approach):** LLM-driven TCG is guided by structured human expertise, a strategy central to our proposed SAGA framework (detailed in Section 4).

### 3.1 Problem Definition

Formally, for a programming problem $P \in \mathcal{P}$ with description $D$, input space $\mathcal{X}_P$, and ground-truth solution $f_P : \mathcal{X}_P \to \mathcal{Y}_P$, a TCG method aims to produce a test case set $T = \{(I_i, O_i)\}_{i=1}^n$, where each input $I_i \in \mathcal{X}_P$ and its corresponding output $O_i = f_P(I_i)$. To quantify the quality of such generated test suites, we employ two key metrics:

**Definition 1** (Detection Rate (DR)). *A **solution-level** metric that answers the question: "For a specific buggy program, can our test suite find at least one bug?" Mathematically, DR measures a test suite's ability to detect an error in a single incorrect solution $S \neq f_P$. It is the probability that at least one test case in $T$ causes $S$ to fail. Let $E_i$ be the event that $S$ fails on test case $i$. The detection rate is the probability of the union of these events: $\epsilon_S(T) = \mathbb{P}(\bigcup_{i=1}^n E_i)$. The final reported DR for a TCG method is the average of these binary outcomes (1 if a bug is found, 0 otherwise) across all known incorrect solutions for a given problem set.*

**Definition 2** (Verifier Accuracy (VAcc)). *A stricter, **problem-level** metric that answers the question: "For a given problem, can our test suite find bugs in all known incorrect solutions?" VAcc evaluates the systematic completeness of the verifier. It is a binary indicator that equals 1 if and only if the test suite $T$ successfully detects a failure in every incorrect solution within the set $\mathcal{S}_{wrong}(P)$. Formally: $\mathrm{VAcc}(T) = \mathbb{I}(\forall S \in \mathcal{S}_{wrong}(P), \epsilon_S(T) > 0)$, where $\mathbb{I}(\cdot)$ is the indicator function. $\mathrm{VAcc}(T) = 1$ signifies that the verifier is robust enough to reject all known faulty programs for that problem.*

### 3.2 Investigating Current TCG Paradigms and Their Limitations

To ground our exploration of TCG effectiveness, we leverage **TCGBench**, a dataset we curated comprising 1840 recent programming problems from Atcoder, Codeforces, and Nowcoder, along with an average of 36.66 incorrect user submissions per problem. This rich resource (detailed in Appendix D) facilitates rigorous evaluation of TCG methodologies. Our analysis employ open-source LLMs: DeepSeek-V3-0324, Qwen2.5-72B-Instruct, and Qwen2.5-Coder-32B-Instruct with the greedy decoding strategy.

#### For paradigm 1: Is Direct LLM-based Test Case Generation Effective?

Directly prompting an LLM to generate complete test cases (inputs $I_i^{\mathrm{LLM}}$, outputs $O_i^{\mathrm{LLM}}$) from a problem description $P$, depends heavily on the LLM's deep comprehension, particularly of edge cases. Our experiments (Figure 3), involving LLM generation of 50 diverse test cases per problem,

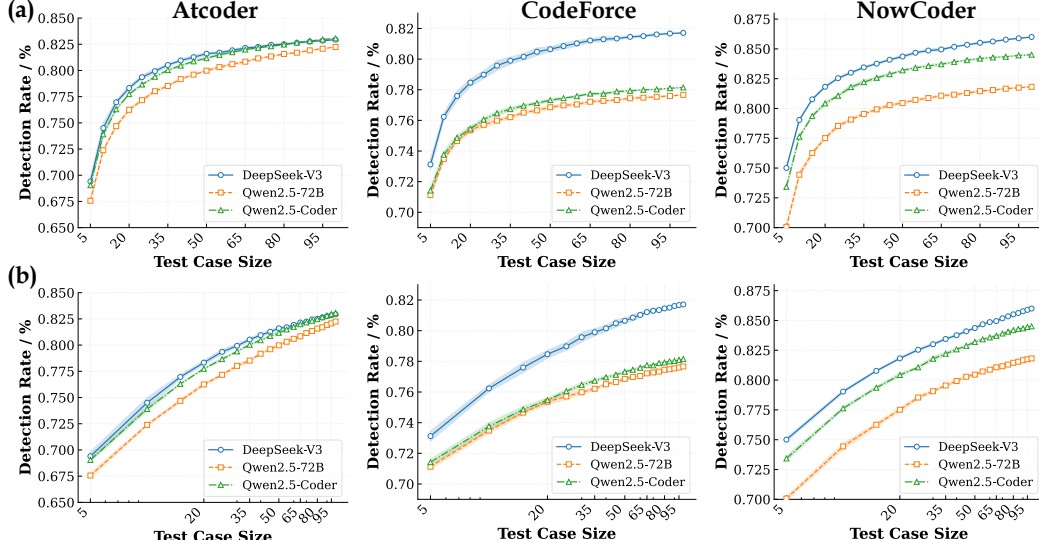

Figure 4: Experimental validation of Input-Interpreter (random sampling) limitations on TCGBench. (a) Detection rate vs. number of test cases (linear scale), showing clear saturation below 100%. (b) Detection rate vs. log of the number of test cases (semi-log scale), illustrating diminishing returns consistent with the theoretical upper bound $1 - (1 - \bar{p})^{n_{\text{eff}}}$, validating the impact of correlation $\bar{\rho}$.

reveal the issues. The *retention rate* (proportion of valid tests post-verification against ground truth) is very low, indicating unreliable quality. This results in poor overall DR (often <60%) and VAcc (<10%). Moreover, LLM-generated solutions easily pass these self-generated tests. Notably, on AtCoder problems with historical official tests[4], LLM solutions performed substantially better on their own generated tests, suggesting such tests fail to challenge the model's cognitive biases.

**For paradigm 2: Can a Large Number of Inputs from an Input-Interpreter Approach Compensate for Low Quality?**

The Input-Interpreter paradigm, where an LLM generates random inputs $I_i^{\text{Rand}} \sim \mathcal{X}_P$ for a ground-truth interpreter, is employed by benchmarks like LiveCodeBench [20]. While generating many tests is feasible, merely increasing the quantity $n$ does not fundamentally improve the detection rate. This limitation arises from inherent correlations between test cases. We propose here a corollary for the upper bound of the detection rate[5]:

> **Corollary 1** (Asymptotic Saturation of Detection Rate). *As the number of generated test cases $n \to \infty$, if $\bar{p}$ (the average probability of a single test detecting an error, $0 < \bar{p} < 1$) and $\bar{\rho}_{\text{eff}}$ (the effective average positive correlation between detection events, $\bar{\rho}_{\text{eff}} > 0$) are stable characteristics, the approximate upper bound on the detection rate $\epsilon_S(T)$ converges to:* $\lim_{n \to \infty} \epsilon_S(T) \approx 1 - (1 - \bar{p})^{1/\bar{\rho}_{\text{eff}}} < 1$.

This corollary implies that due to inter-test correlation $\bar{\rho}_{\text{eff}}$, simply increasing the number of random tests cannot guarantee 100% error detection; the detection rate will saturate. The underlying reasoning involves the concept of an "effective sample size" $n_{\text{eff}} \approx n/(1 + (n-1)\bar{\rho}_{\text{eff}})$ [21], which quantifies the diminishing utility of additional correlated tests. Our experiments on TCGBench (Figure 4) confirm this: detection rates plateau (Fig. 4(a)), and marginal gains diminish rapidly. Notably, plotting the detection rate against the logarithm of the number of test cases (Fig. 4(b)) reveals a trend consistent with our derived theoretical upper bound, further validating the impact of correlation $\bar{\rho}$.

Beyond DR and Acc, to more deeply quantify the intrinsic quality and efficiency of TCG strategies like SAGA (introduced in Section 4), we employ two advanced metrics. These provide observable insights into test suite characteristics related to $\bar{p}$ (average test potency) and $\bar{\rho}_{\text{eff}}$ (inter-test correlation):

---

[4]Early AtCoder problems provided official test cases; however, from December 2024 onwards, test cases are no longer publicly available for new contests.

[5]The full derivation is provided in Appendix C.

- **Distinct Error Pattern Coverage (DEPC)**: For a test suite $\mathcal{T}$ and $N_P$ problems, let $v(t_k)$ be the error pattern vector for test $t_k$. DEPC is $\left| \left\{ v(t_k) \mid t_k \in \mathcal{T} \text{ and } \|v(t_k)\|_1 \geq 1 \right\} \right|$. The **Diversity Ratio** is $\text{DEPC}(\mathcal{T})/n$. Higher DEPC suggests lower $\bar{\rho}_{\text{eff}}$, indicating broader unique error detection.
- **Normalized Area Under the Accuracy-Number of test cases Curve (AUC-AccN)**: For Verifier Accuracy $Acc(k)$ with $k$ tests up to $N$,

$$\text{AUC@N} \approx \frac{1}{N - k_{min}} \sum_{i=k_{min}}^{N-1} \frac{Acc(k_i) + Acc(k_{i+1})}{2}(k_{i+1} - k_i).$$

Higher AUC@$N$ indicates superior average verifier accuracy, reflecting potent (high $\bar{p}$) and efficiently diverse tests.

Detailed explanations and derivations for these metrics are in Appendix B.

## 4 SAGA: A Human-LLM Collaborative Framework for Advanced TCG

The preceding analysis in Section 3.2 revealed critical limitations in prevalent Test Case Generation (TCG) paradigms. As demonstrated, Direct Generation suffers from low test quality (Figure 3), while the Input-Interpreter approach is constrained by test case homogeneity and saturating performance returns (Figure 4). To address these shortcomings, we introduce **SAGA** (**S**trategic **A**dversarial & Constraint-differential **G**ener**A**tive workflow), a novel human-LLM collaborative framework (Figure 5). SAGA systematically generates high-quality, diverse, and discriminative test suites by maximizing test potency ($\bar{p}$) and diversity (lowering correlation $\bar{\rho}$). Additionally, we trained **TCGCoder-7B**, a SAGA-distilled 7B specialist model from 15,000 problems (details in Appendix J), as a strong TCG baseline and reference.

### 4.1 The SAGA Framework: Integrating Human Expertise

Recognizing the limitations of naive TCG, SAGA explores the integration of human expertise. Intuitively, leveraging human problem-solving insights should enhance TCG. Table 1 shows that incorporating simple human priors (e.g., boundary values from $\mathcal{S}_{\text{human}}$) into basic TCG paradigms on TCGBench-Full yields some improvement. However, these gains are often marginal, failing to fully exploit LLM potential for detecting complex flaws. While methods like EvalPlus [28] use human solutions, it's often for mutating seeds or formatting inputs, not deeply guiding large-scale challenging test generation.

Table 1: Initial Impact of Incorporating Simple Human Priors ($\mathcal{S}_{\text{human}}$) into Basic TCG Paradigms on AtCoder Results (Accuracy@50).

| Method | Accuracy@50 |
|---|---|
| Direct Gen. (Paradigm 1) | 11.30% |
| Direct Gen. + Simple Priors | **15.06%** (+3.76%) |
| Input-Interpreter (Paradigm 2) | 23.36% |
| Input-Interpreter + Simple Priors | **27.95%** (+4.59%) |

SAGA advances beyond such superficial integration. As depicted in Figure 5, it deeply incorporates multifaceted human programming insights—from both correct solutions (GroundTruth, $\mathcal{S}_{\text{human}}$) and incorrect submissions (Human Bugs, $\mathcal{S}_{\text{wrong}}$)—with LLM reasoning via a structured, dual-pronged analytical strategy. An LLM is fed the problem description $P$ and insights gleaned from human solutions through a customized prompting module. This process generates Python *Case Scripts* to produce test inputs, accompanied by *Math Explanations* and *Self-Validation* code to ensure correctness and relevance. The generated inputs are then processed by an *Interpreter* (ground-truth solution) to yield test outputs, forming the final test cases. SAGA's core analytical dimensions are:

- **Multidimensional Analysis (Leveraging $\mathcal{S}_{\text{human}}$):** This dimension extracts profound insights from correct solutions to engineer challenging tests. It involves:
  1. *Constraint Handling Differences:* Discrepancies in how $S_{\text{wrong}}$ and $S'_{\text{correct}}$ manage problem-specific constraints.
  2. *Defense Pattern Deconstruction:* This is where the *Math Explanation* component (Figure 5) plays a key role. It transforms the implicit programming logic from human code into explicit, structured test instructions. Diverse defensive logic and problem-solving strategies within $\mathcal{S}_{\text{human}}$ are decomposed into formal mathematical or logical constraints (e.g., "equivalence class: player pairs", "boundary_value: [(1,2), (N,N-1)]"). This process allows SAGA to purposefully

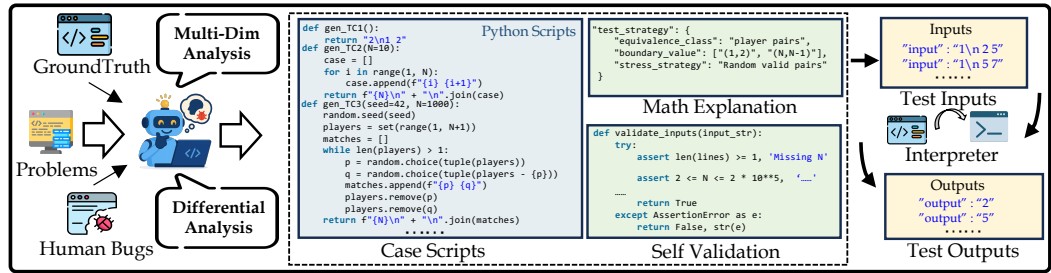

Figure 5: Overview of the SAGA framework. SAGA leverages both GroundTruth (correct human solutions) and Human Bugs (incorrect submissions) alongside the Problem description. An LLM performs Multi-Dimensional Analysis and Differential Analysis to generate Python Case Scripts for test input synthesis. These scripts are accompanied by Math Explanations (capturing testing strategies and constraints) and Self-Validation code. The generated Test Inputs are then passed to an Interpreter (ground-truth solution) to produce Test Outputs, forming the final test cases.

  target the core mathematical and logical properties of the problem—such as singularities, extremal values, or specific structural properties—guiding the generation of meaningful edge and adversarial test cases that simple random generation would miss.
  3. *Targeted Test Generation:* Using these constraints and pitfalls to guide the LLM in constructing challenging test inputs $I$ via the *Case Scripts*.
This analytical approach aims to generate test cases that not only cover a wide spectrum of valid scenarios, including complex boundary conditions and intricate interactions, but also enhance test diversity (lowering $\bar{\rho}$) and individual test potency (increasing $\bar{p}$).

- **Differential Analysis (Leveraging $\mathcal{S}_{\text{wrong}}$):** This addresses error types missed by analyzing only correct solutions. It compares failed submissions ($\mathcal{S}_{\text{wrong}}$) with their corrected versions ($S'_{\text{correct}}$) to find inputs $I_{\text{diff}}$ where $S_{\text{wrong}}(I_{\text{diff}}) \neq S'_{\text{correct}}(I_{\text{diff}})$, revealing common error patterns. This targets:
  1. *Constraint Handling Differences:* Discrepancies in how $S_{\text{wrong}}$ and $S'_{\text{correct}}$ manage problem-specific constraints.
  2. *Lack of Defensive Completeness:* Deficiencies in $S_{\text{wrong}}$ related to handling edge cases or boundary inputs, as revealed by comparison with $S'_{\text{correct}}$.
  3. *Failure Pattern Analysis:* Generating specific inputs that trigger failures in $S_{\text{wrong}}$ but are correctly handled by $S'_{\text{correct}}$.
The incorporation of these differentially identified inputs $I_{\text{diff}}$ into the test suite $T$ creates a more rigorous and challenging evaluation framework, as it specifically targets known failure modes, thereby substantially increasing the discriminative power of the resulting verifier.

The *Self-Validation* scripts (Figure 5) ensure that generated test inputs adhere to problem constraints and the intended testing strategy before execution. For Multidimensional Analysis, SAGA leverages insights from 10 distinct, correct user solutions per problem to broaden perspective. In Differential Analysis, it meticulously pairs a user's correct solution with their most recent preceding incorrect submission. This focus on closely related yet differing attempts enhances SAGA's ability to identify subtle error patterns and generate challenging corner cases. By synergistically combining these refined analytical dimensions, SAGA produces comprehensive and highly effective test suites.

It is important to note that SAGA is designed for efficiency. The main computational cost lies in a one-time analysis phase per problem, where the LLM performs a single, powerful analysis of human solutions to generate a Python *Case Script*. Crucially, it is this generated script—not the LLM—that then programmatically generates a large volume of test inputs. This design avoids the high cost and latency of repeatedly invoking the LLM for every single test case, making the framework scalable.

## 4.2  Experimental Validation of SAGA

SAGA's effectiveness was initially validated on the **TCGBench**. Figure 6 visually summarizes SAGA's multifaceted superiority in this context, comparing it against the random Input-Interpreter baseline and its core analytical components (Multidimensional Analysis and Differential Analysis).

**Key Findings from Comparison on TCGBench (Fig. 6):** SAGA markedly improves *solution vetting capabilities*: its DR surpasses 93.81% (vs. baseline's 82.85% at $n = 100$) and VAcc reaches

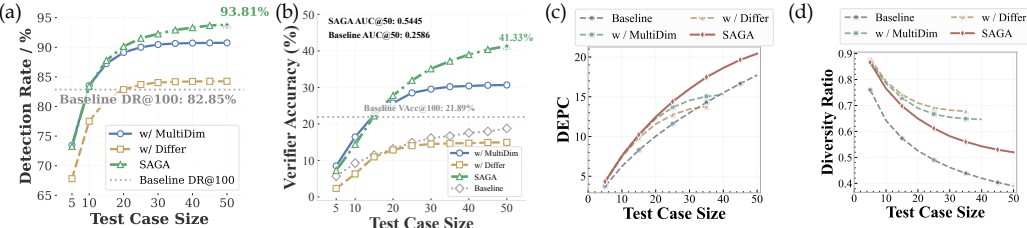

Figure 6: SAGA outperforms the Baseline (Random Input-Interpreter) and its individual analytical components (Multidimensional Analysis leveraging $\mathcal{S}_{\text{human}}$; Differential Analysis leveraging $\mathcal{S}_{\text{wrong}}$) on the AtCoder subset of the full **TCGBench** across: (a) Detection Rate, (b) Verifier Accuracy (with AUC@50 values), (c) Distinct Error Pattern Coverage (DEPC), and (d) Diversity Ratio. Dotted lines in (a) & (b) show baseline performance at $n = 100$. See Appendix F for SAGA's performance on other platforms within the full TCGBench.

41.33% (vs. baseline's 21.89% at $n = 100$) with only 50 tests. SAGA's AUC@50 (0.5445) more than doubles the baseline's (0.2586), showcasing superior efficiency. SAGA also generates test suites of *superior intrinsic quality*: it achieves the highest DEPC (broader error coverage, lower $\bar{\rho}$) and Diversity Ratio (more efficient error discovery per test). Both Multidimensional and Differential analysis components individually outperform the baseline, but their synergy in SAGA yields optimal results. This robustly validates SAGA's systematic leveraging of human insights on a large scale.

### 4.2.1   Main Results and Analysis on TCGBench-Lite

For focused main comparisons and ablation studies, we curated **TCGBench-Lite**, a challenging subset of 270 problems from AtCoder, Codeforces, and Nowcoder contests since **June 2024**. This ensures contemporary relevance and minimizes potential data leakage. To further guarantee the impartiality of our results, particularly for the evaluation of our SAGA-distilled model **TCGCoder-7B**, we employed a strict chronological split: the training data for TCGCoder-7B was sourced entirely from problems published before 2023. This temporal separation ensures the model is evaluated on genuinely unseen problems. TCGBench-Lite includes an average of 41.41 incorrect submissions ($\mathcal{S}_{\text{wrong}}$) per problem. Its difficulty distribution (Easy: 27.04%, Medium: 32.59%, Hard: 40.37%) was determined by platform tags and contest characteristics (details in Appendix E). This curated set allows for rigorous yet manageable evaluation. Unless specified, all LLM-driven methods use DeepSeek-V3-0324 as the backbone for fair comparison. We evaluate against the **Input-Interpreter (LiveCodeBench-style random sampling)**, **TestChain** [23], **EvalPlus** [28], and our SAGA-distilled **TCGCoder-7B**. Results are in Table 2.

**Analysis of Main Comparison (Table 2):** SAGA's structured integration of human insights yields substantial gains over other TCG methods on TCGBench-Lite. Notably, SAGA achieves an AUC@50 of 0.2228, surpassing the Input-Interpreter (0.1234), which is limited by random sampling's homogeneity. This highlights SAGA's ability to produce more consistently effective tests. While EvalPlus achieves a high raw Diversity Ratio through mutation, its lower AUC@50 (0.1278) suggests that SAGA's deep analysis of human solutions is more critical for overall verifier quality than mere test variety. TestChain, lacking rich human priors, performs weakest. Crucially, our SAGA-distilled **TCGCoder-7B** (AUC@50: 0.1890) outperforms all these established baselines, even when they utilize a much larger backbone model (DeepSeek-V3-0324). This demonstrates SAGA's potential to distill effective TCG strategies into smaller, specialized, and highly capable models, thereby making advanced TCG more accessible.

**Analysis of Ablation Studies (Table 2):** *Value of Structured Human Insights:* Isolating SAGA's Multidimensional Analysis (leveraging $\mathcal{S}_{\text{human}}$) and Differential Analysis (leveraging $\mathcal{S}_{\text{wrong}}$) reveals that both significantly outperform simpler prompting (SimpleCOT) and the Input-Interpreter baseline in terms of AUC@50. This underscores that structured analysis of human solutions, whether correct or incorrect, is fundamental for higher-quality verifiers. The full SAGA framework, synergistically combining both analytical dimensions, achieves the optimal overall quality (highest AUC@50), confirming their complementary benefits. *Importance of Prompt Design:* The substantial performance degradation when replacing SAGA's detailed, insight-driven prompts with generic CoT (SimpleCOT) highlights that the *methodology* of integrating human insights is as critical as the insights themselves

Table 2: Experimental Results: Main Comparison of SAGA with Baselines and Ablation Studies on TCGBench-Lite. Metrics are DR@$k$, VAcc@$k$, AUC@50, and DivRatio@50. All methods use DeepSeek-V3-0324 backbone unless specified in ablation.

| Method / Configuration | DR@20 | DR@50 | VAcc@20 | VAcc@50 | AUC@50 | DivRatio@50 |
|---|---|---|---|---|---|---|
| *Main Comparison with Baseline TCG Methods* | | | | | | |
| TestChain [23] | 65.91% | 68.31% | 8.12% | 11.88% | 0.0841 | 50.09% |
| Input-Interpreter (LiveCodeBench-style [20]) | 77.84% | 81.07% | 12.36% | 16.72% | 0.1234 | 79.42% |
| EvalPlus [28] | 67.52% | 71.12% | 11.56% | 15.15% | 0.1139 | 79.27% |
| TCGCoder-7B (SAGA-distilled model) | 85.14% | 89.44% | 17.93% | 29.11% | 0.1890 | 94.43% |
| **SAGA (DeepSeek-V3 Backbone)** | 85.66% | **90.62%** | **22.40%** | 32.58% | **0.2228** | 94.06% |
| | | | | | | |
| *SAGA Ablation Studies* | | | | | | |
| *Analytical Component Ablation* | | | | | | |
| SAGA w/ Multidim. Analysis only | 84.51% | 88.00% | 20.70% | 26.05% | 0.1923 | 95.81% |
| SAGA w/ Differential Analysis only | 84.31% | 88.16% | 19.85% | 26.67% | 0.1926 | 94.41% |
| | | | | | | |
| *Prompt Design Ablation* | | | | | | |
| SimpleCOT Prompt for SAGA | 83.36% | 84.54% | 15.61% | 19.11% | 0.1424 | 96.23% |
| Random Input w/ GT for SAGA | 82.31% | 86.64% | 16.44% | 22.70% | 0.1616 | 85.38% |
| EvalPlus w/ GT for SAGA | 76.72% | 79.56% | 11.67% | 20.44% | 0.1278 | 89.49% |
| | | | | | | |
| *Base LLM Ablation for SAGA* | | | | | | |
| SAGA w/ Qwen2.5-Coder-7B-Instruct | 78.88% | 79.78% | 19.70% | 22.96% | 0.1810 | **96.80%** |
| SAGA w/ Qwen2.5-72B-Instruct | 82.77% | 85.08% | 20.30% | 26.46% | 0.1943 | 94.92% |
| SAGA w/ Qwen2.5-Coder-32B-Instruct | **86.25%** | 90.54% | 20.74% | **32.73%** | 0.2139 | 94.72% |

for effective TCG. *Robustness Across LLMs:* SAGA demonstrates commendable robustness when paired with different LLM backbones. While peak performance on specific metrics can vary with the LLM (e.g., SAGA with Qwen2.5-Coder-32B-Instruct yields the highest VAcc@50), the SAGA framework with its default DeepSeek-V3 backbone consistently provides the best overall AUC@50. This indicates SAGA's comprehensive design is more impactful for holistic verifier quality than relying on specific LLM capabilities or optimizing for isolated metrics like Diversity Ratio. Notably, SAGA also shows strong performance with Qwen2.5-Coder-7B-Instruct (TCGCoder-7B's base model), further validating SAGA's efficacy on smaller, specialized coder models.

# 5   Towards Advanced Applications of SAGA

SAGA's proven capability in generating high-quality, diverse, and discriminative test suites (Section 4) enables advancements in LLM code evaluation and training. This section primarily explores the development of a superior code generation benchmark, **CodeComPass**, while also noting the potential for SAGA-enhanced verifiers to contribute to more reliable Reinforcement Learning from Verifiable Rewards (RLVR).

**CodeCompass: A SAGA-Enhanced Benchmark for Code Generation Evaluation**

To address the need for more challenging and discerning evaluation of LLM code generation models, we introduce **CodeCompass**. The verifiers within CodeCompass are synthesized using SAGA for the 270 contemporary problems that also constitute TCGBench-Lite (detailed in Appendix E), ensuring relevance and minimizing data leakage risks. Each problem in CodeCompass features a rich test suite, averaging 50.54 SAGA-generated test cases, meticulously curated to ensure comprehensive coverage (with additional manual curation applied if initial generation yielded fewer than a target threshold).

For comparative analysis of verifier quality and impact on model evaluation, we focus on a shared subset of 101 AtCoder problems that overlap between our CodeCompass verifiers and the test suites used by LiveCodeBench-v6. This allows for a direct comparison on common ground.

**Superior Verifier Quality of CodeCompass.** We first establish the intrinsic quality of CodeCompass verifiers using core TCG metrics against those of LiveCodeBench-v6 on the shared AtCoder subset (Table 3). CodeCompass demonstrates markedly higher efficacy in identifying faulty solutions, greater efficiency in discovering unique error patterns (Diversity Ratio@40: +43.13%), and faster convergence to quality. This confirms that SAGA produces verifiers that are intrinsically more diverse, discriminative, and effective.

Table 3: Verifier Quality: CodeCompass vs. LCB-v6 (Shared Subset, @40 tests).

| Metric | LiveCodeBench-v6 | CodeCompass |
|---|---|---|
| DR@40 | 78.85% | **93.44%** |
| VAcc@40 | 19.61% | **30.39%** |
| DivRatio@40 | 53.56% | **96.69%** |
| AUC@40 | 0.1388 | **0.1980** |

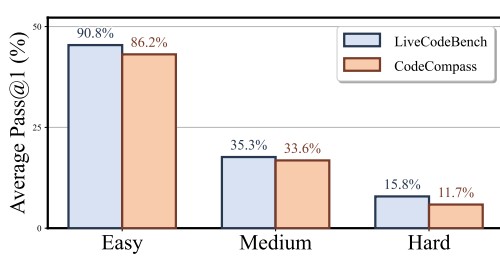

Figure 7: Avg. Pass@1 by Difficulty.

Figure 8: Model Ranking Changes.

**Enhanced Discriminative Power for Code Generation Model Evaluation.** This superior verifier quality translates directly to a more challenging and discerning evaluation for LLM code generation models. As shown in Figure 7, CodeComPass consistently elicits lower average Pass@1 rates across all problem difficulties compared to LiveCodeBench-v6 on the shared subset, indicating a more rigorous test. Critically, this increased stringency enhances discriminative power (Figure 8): when evaluated on CodeComPass, the average Pass@1 for various models drops by a relative 9.56% compared to their performance on LiveCodeBench-v6. This relative decrease leads to a re-ranking of models (e.g., Qwen2.5-72B and Qwen2.5-Coder-32B switch relative positions). This ability to expose nuanced differences in model capabilities confirms that CodeComPass offers a more robust and insightful assessment of true code generation proficiency.

Meanwhile, the superior SAGA-generated verifiers, exemplified by CodeComPass, promise to enhance Reinforcement Learning from Verifiable Rewards (RLVR) frameworks [8, 10] by delivering more accurate reward signals and mitigating reward hacking, thereby fostering more robust and capable code generation models.

# 6 Conclusion

This paper critically re-evaluates LLM-based Test Case Generation (TCG), highlighting current verifier limitations and formalizing key quality metrics alongside TCGBench, a foundational TCG research dataset. We introduce **SAGA**, a novel human-LLM collaborative framework that integrates human programming insights with LLM reasoning to demonstrably generate superior test suites. Leveraging SAGA, we developed **CodeComPass**, an enhanced verifier suite for robust code generation evaluation, and distilled **TCGCoder-7B**, a specialized, efficient TCG model. Collectively, these contributions significantly advance reliable LLM code evaluation and pave the way for future automated test synthesis and effective adversarial testing.

# 7 Acknowledgment

This work was supported by the National Nature Science Foundation of China (No. 62192781, No. 62272374), the Natural Science Foundation of Shaanxi Province (2024JC-JCQN-62), the State Key Laboratory of Communication Content Cognition under Grant No. A202502, the Key Research and Development Project in Shaanxi Province (No. 2023GXLH-024), Shanghai Oriental Talents Project BJZH2024070. We also thank the OpenCompass team at Shanghai AI Laboratory and our colleagues for their support and discussions.

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

# Appendix

## A    Related Works

### A.1    Advancements and Challenges in LLM-based Code Generation and Evaluation

Large Language Models (LLMs) have revolutionized automated code generation, with models like AlphaCode [26], GPT-4 [37], Gemini [2], and specialized code models such as CodeT5 [44, 45], StarCoder [24], DeepSeek Coder [13], CodeLlama [40], and Qwen2.5-Coder [46, 18] demonstrating remarkable capabilities. These models, whether trained via Supervised Fine-Tuning or enhanced through Reinforcement Learning (RL) with execution feedback (e.g., AlphaCode [26], PPOCoder [42], CodeRL [22]), increasingly match or exceed human performance on diverse programming benchmarks.

The reliable evaluation of these sophisticated models is paramount and hinges on the quality of **Code Verifiers**—typically test suites—that ascertain the functional correctness of generated code. This is especially critical for RL frameworks employing verifiable rewards (RLVR) [8, 10, 36, 17], where test case quality directly impacts reward accuracy and training efficacy. However, as highlighted in Section 1, the comprehensiveness and robustness of verifiers often present a significant bottleneck, underscoring the critical need for effective Test Case Generation.

To this end, the research community has developed numerous benchmarks. Early benchmarks like HumanEval [6] and MBPP [3] provided foundational testbeds. Subsequent efforts like EvalPlus [28, 29] and MBPP-Plus aimed to improve robustness by expanding existing test sets. More comprehensive benchmarks such as TACO [25], CodeContests [26] (focused on algorithmic problems), BigCodeBench [50] (complex multi-library tasks), LiveCodeBench [20] (simulating online judge environments), and HumanEvalPack [49] (multilingual program synthesis) have broadened the evaluative scope. Concurrently, dedicated TCG benchmarks like TESTEVAL [43] and TestGenEval [19] have emerged. Despite these advancements, persistent issues in test case quality, coverage, and potential LLM-centric biases necessitate continuous efforts towards developing high-quality, diverse, and unbiased Code Verifiers.

### A.2    Methodologies for Test Case Generation (TCG)

TCG is fundamental to validating code correctness and providing feedback for both evaluation and training of code generation models. While formally defined as a distinct task in this paper, TCG principles are integral to benchmark creation (e.g., LiveCodeBench [20], TESTEVAL [43]) and RL data pipelines (e.g., ACECODER [48], CodeRL [22]). However, suboptimal test quality in existing datasets (e.g., TACO [25], CodeForces-CoTs [39]) can lead to overestimated model performance and reward hacking, particularly in rule-based RL systems like AlphaCode [26] and DeepSeek Coder [13], thus highlighting the urgent need for effective TCG.

**Traditional TCG Techniques:** Software testing has a rich history of TCG. **Search-Based Software Testing (SBST)** [1, 14] uses metaheuristics like genetic algorithms [4, 35] to optimize for coverage criteria, though it can be computationally intensive. **Symbolic Execution** [9, 5] explores program paths systematically but faces path explosion and challenges with complex code. **Fuzzing** [34, 11], especially coverage-guided variants like AFL [11] and libFuzzer [41], excels at finding crashes and vulnerabilities by mutating inputs, though it may lack semantic depth for logical tests. Feedback-directed random testing [47, 38] also discovers defects but can sometimes lack diversity.

**LLM-based TCG Paradigms:** With the rise of LLMs, new TCG methods have emerged, leveraging semantic understanding for potentially greater coverage and diversity. As discussed in Section 3, these generally follow two paradigms:

1. *Direct Generation:* LLMs produce complete test cases (inputs and outputs). This includes assertion-focused methods (e.g., CodeCoT [15], AgentCoder [16]) and direct input-output synthesis (e.g., TestChain [23], CodeRM [33], AceCoder [48]), aiming for logical coverage and boundary conditions.
2. *Input-Interpreter:* LLMs generate test inputs, which are then executed by a ground-truth solution to derive outputs. This is seen in LiveCodeBench [20],CodeForce-Cot [39] and related to LLM-

guided fuzzing. EvalPlus [28, 29] also aligns by mutating seed inputs for execution. TestChain [23] also showed improvements by decoupling input and output generation.

**Advanced and Specialized TCG Approaches:** In RL contexts, dynamic TCG is crucial. CodeRM-8B [33] adjusts test quantity by problem difficulty. LEVER [36] uses learned verifiers, and other works focus on execution-feedback for reward model optimization, all highlighting the impact of test quality on RL outcomes. Differential testing, as in AID [30], compares program versions to expose bugs, showing strong results on datasets like TrickyBugs [30] and EvalPlus [28]. Recent efforts also explore generating tests targeting prior failures [7] or using LLMs for test suite refinement.

Despite these diverse approaches, achieving comprehensive logical coverage, generating truly diverse and challenging corner-case tests, and maintaining computational efficiency remain significant hurdles. Furthermore, many LLM-centric TCG methods may perpetuate biases inherent in the LLMs themselves (Section 1). This paper's SAGA framework addresses these limitations by proposing a novel human-LLM collaborative paradigm that systematically integrates deep human insights from both correct and incorrect solutions. By focusing on a dynamic, adaptive, and efficient TCG process, SAGA aims to enhance the reliability of LLM code evaluation and training, paving a new path for TCG research.

# B  Formulation and Interpretation of Advanced Evaluation Metrics

This section provides detailed explanations and interpretations for the advanced metrics introduced in Section 3.2 to evaluate the intrinsic quality of test suites.

## B.1  Distinct Error Pattern Coverage (DEPC) and Diversity Ratio

**Formulation (from main text):** For a test suite $\mathcal{T}$ and $N_P$ problems, let $v(t_k)$ be the $N_P$-dimensional binary error pattern vector for test $t_k$ (where $v(t_k)_j = 1$ if $t_k$ reveals an error for problem $j$).

$$\text{DEPC}(\mathcal{T}) = \left| \left\{ v(t_k) \mid t_k \in \mathcal{T} \text{ and } \|v(t_k)\|_1 \geq 1 \right\} \right|.$$

The **Diversity Ratio** is $\text{DEPC}(\mathcal{T})/n$, where $n = |\mathcal{T}|$.

**Interpretation:** DEPC measures the breadth of error coverage by counting the number of unique ways (patterns) in which the test suite can detect failures across a set of problems. A higher DEPC signifies that the test suite is capable of identifying a wider variety of distinct error types or combinations of errors. This directly relates to the concept of inter-test case correlation ($\bar{\rho}_{\text{eff}}$) discussed in our theoretical model (Appendix C): a test suite with high DEPC is likely composed of tests that are less correlated in terms of the errors they detect, thus having a lower $\bar{\rho}_{\text{eff}}$. The Diversity Ratio normalizes DEPC by the number of test cases, indicating the average efficiency of each test in contributing a new, distinct error pattern. A high Diversity Ratio suggests that the test suite is not only diverse but also concise, with less redundancy among its test cases.

## B.2  Normalized Area Under the Accuracy-Number of test cases Curve (AUC-AccN)

**Formulation (from main text):** For Verifier Accuracy $Acc(k)$ achieved with a test suite of size $k$ (up to a maximum $N$, starting from $k_{min}$), the AUC-AccN is approximated by the trapezoidal rule:

$$\text{AUC@N} \approx \frac{1}{N - k_{min}} \sum_{i=k_{min}}^{N-1} \frac{Acc(k_i) + Acc(k_{i+1})}{2} (k_{i+1} - k_i).$$

**Interpretation:** AUC-AccN quantifies the average Verifier Accuracy as the test suite size grows, providing a single scalar value to compare the overall effectiveness and efficiency of different TCG strategies. A higher AUC@$N$ (ranging from 0 to 1) indicates that a TCG strategy consistently generates test suites that achieve higher verifier accuracy across various sizes up to $N$. This metric is a composite reflection of both the average potency of individual test cases ($\bar{p}$) and the effective diversity of the test suite (related to $\bar{\rho}_{\text{eff}}$).

- **Impact of $\bar{p}$ (Test Potency):** A higher average $\bar{p}$ means individual tests are more likely to detect errors. This leads to a steeper initial rise in the $Acc(k)$ curve and a higher overall level of accuracy

achieved, both of which contribute to a larger AUC@$N$. Thus, AUC-AccN is particularly sensitive to $\bar{p}$.

- **Impact of Diversity (low $\bar{\rho}_{\text{eff}}$):** Greater diversity (lower $\bar{\rho}_{\text{eff}}$), empirically reflected by a high DEPC, allows the $Acc(k)$ curve to sustain its rise or plateau at a higher level for a larger number of test cases before saturation effects become dominant. This sustained high accuracy also contributes to a larger AUC@$N$.

Therefore, a high AUC@$N$ signifies a TCG strategy that excels at generating tests that are individually powerful (high $\bar{p}$) and collectively non-redundant (low $\bar{\rho}_{\text{eff}}$), leading to efficient and robust verifier construction within the specified test suite size limit. It reflects a better overall quality of the generated test cases in achieving high average performance.

## C   Theoretical Analysis of Detection Rate

We analyze the detection rate $\epsilon_S(T) = \mathbb{P}(X \geq 1)$ for a test suite $T = \{t_1, \ldots, t_n\}$, where $X = \sum_{i=1}^{n} \mathbf{1}_{E_i}$ and $p_i = \mathbb{P}(E_i)$ is the probability that test $t_i$ detects an error.

### C.1   Modeling Correlated Heterogeneous Bernoulli Trials

The expectation of $X$ is $\mathbb{E}[X] = \sum p_i = n\bar{p}$, where $\bar{p} = \frac{1}{n}\sum p_i$. The variance is $\text{Var}(X) = \sum p_i(1-p_i) + \sum_{i \neq j} \text{Cov}(\mathbf{1}_{E_i}, \mathbf{1}_{E_j})$.

To model the effect of correlation in a tractable manner that allows for insights similar to the homogeneous case (uniform $p$, uniform $\rho$), we consider an *approximating model*. We define an effective average pairwise covariance, $\bar{C}_{\text{eff}}$, and an average individual variance, $\bar{\sigma}_p^2 = \frac{1}{n}\sum p_i(1 - p_i)$. We then model the variance as if it arose from a system with these averaged second-order characteristics:
$$\text{Var}_{\text{approx}}(X) = n\bar{\sigma}_p^2 + n(n-1)\bar{C}_{\text{eff}}.$$
If we further posit that these average characteristics can be related through an effective average correlation $\bar{\rho}_{\text{eff}}$ such that $\bar{C}_{\text{eff}} \approx \bar{\rho}_{\text{eff}}\sqrt{\text{avg}(p_i(1-p_i))^2}$ or, more simply for conceptual linkage, $\bar{C}_{\text{eff}} \approx \bar{\rho}_{\text{eff}}\bar{p}(1-\bar{p})$ (assuming $p_i$ are not excessively dispersed, making $\bar{\sigma}_p^2 \approx \bar{p}(1-\bar{p})$), then:
$$\text{Var}_{\text{approx}}(X) \approx n\bar{p}(1-\bar{p})[1 + (n-1)\bar{\rho}_{\text{eff}}]. \tag{1}$$
This equation models the variance of $X$ as if it were a sum of $n$ Bernoulli trials with a common success probability $\bar{p}$ and a common pairwise correlation $\bar{\rho}_{\text{eff}}$. The validity of this approximation depends on the actual distribution of $p_i$ and the structure of covariances. However, it serves as a useful model to understand the qualitative impact of average correlation.

**Definition 3** (Model-Based Effective Sample Size $n_{\text{eff}}^*$). *Within this approximating model (Eq. 1), and by analogy to Kish's design effect [21] (where deff* $\approx 1 + (n-1)\bar{\rho}_{\text{eff}}$), the model-based effective sample size is:*
$$n_{\text{eff}}^* \approx \frac{n}{1 + (n-1)\bar{\rho}_{\text{eff}}}.$$
*This $n_{\text{eff}}^*$ represents the number of hypothetical independent Bernoulli($\bar{p}$) trials that would exhibit the variance given by Eq. 1. This is consistent with adjustments for correlated data [32, 12, 27].*

### C.2   Upper Bound and Saturation within the Model

Using $n_{\text{eff}}^*$ and $\bar{p}$ from our model, we analyze the detection rate $\epsilon_S(T) = 1 - \mathbb{P}(X = 0)$.

**Theorem 1** (Model-Based Approximate Upper Bound on Detection Rate). *Within the described approximating model, the detection rate $\epsilon_S(T)$ is approximately upper bounded by:*
$$\epsilon_S(T) \approx 1 - (1-\bar{p})^{n_{\text{eff}}^*} \approx 1 - (1-\bar{p})^{\frac{n}{1+(n-1)\bar{\rho}_{\text{eff}}}}.$$

*Proof Sketch.* The probability $\mathbb{P}(X = 0)$ is approximated by that of $n_{\text{eff}}^*$ independent Bernoulli($\bar{p}$) trials, which is $(1-\bar{p})^{n_{\text{eff}}^*}$. $\square$

This theorem suggests that even when individual $p_i$ vary, if there's an effective positive average correlation $\bar{\rho}_{\text{eff}} > 0$, the system behaves as if it has fewer independent tests, limiting the detection rate.

**C.3 Interpretation and Implications for Performance Metrics**

The derivation above, employing an approximating model based on average parameters ($\bar{p}$, $\bar{\rho}_{\text{eff}}$), robustly indicates that a persistent positive effective correlation among test case detection events leads to a saturation of the overall detection rate. The core insight—that redundancy limits the marginal gain from additional tests—remains. This theoretical observation is crucial for understanding the performance of TCG strategies and their impact on the empirical metrics used in this paper:

- **Verifier Accuracy ($Acc(T)$) and its relation to $\bar{p}$**: The Verifier Accuracy at any given test suite size $n$, $Acc(T_n)$, is fundamentally driven by the test suite's ability to expose errors, which is heavily influenced by the average potency of its constituent test cases. A higher average error detection probability, $\bar{p}$, means that individual tests are, on average, more "powerful" or "incisive." Consequently, a TCG strategy yielding a higher $\bar{p}$ will lead to a verifier that more readily and correctly identifies faulty solutions, directly boosting $Acc(T_n)$. While high correlation ($\bar{\rho}_{\text{eff}}$) can limit the ultimate achievable accuracy by causing early saturation of distinct error discovery, a strong $\bar{p}$ is essential for the accuracy curve to reach a high level in the first place.
- **AUC-AccN as a reflection of sustained high $\bar{p}$ and managed $\bar{\rho}_{\text{eff}}$**: The Area Under the Accuracy-Number of test cases Curve (AUC-AccN), which quantifies the average Verifier Accuracy as test suite size increases, is a composite reflection of both $\bar{p}$ and $\bar{\rho}_{\text{eff}}$. A high $\bar{p}$ ensures that the $Acc(k)$ curve rises steeply and achieves a significant altitude. This initial rapid ascent and the overall height of the curve contribute substantially to a larger AUC-AccN. Concurrently, a lower effective correlation $\bar{\rho}_{\text{eff}}$ (i.e., greater diversity, reflected empirically by DEPC) allows the accuracy to be sustained or to continue growing across a larger number of test cases before significant saturation, thereby expanding the area under the curve. Therefore, strategies achieving a high AUC-AccN are those that likely generate test cases with a consistently high average error detection probability ($\bar{p}$) and effectively manage redundancy (lower $\bar{\rho}_{\text{eff}}$). The magnitude of $\bar{p}$ is particularly critical for the "value" captured by AUC-AccN, as it dictates the average level of accuracy being integrated.
- **DEPC and its relation to $\bar{\rho}_{\text{eff}}$**: DEPC empirically captures the diversity of error patterns. A TCG strategy that yields high DEPC is effectively generating tests with low effective average correlation $\bar{\rho}_{\text{eff}}$, thus mitigating the saturation effect on discovering new types of errors and allowing for a more sustained increase in overall detection capability.

Therefore, the pursuit of TCG methods like SAGA, which aim to enhance both individual test case strength (targeting a higher $\bar{p}$) and inter-test case diversity (targeting a lower $\bar{\rho}_{\text{eff}}$), is theoretically well-founded for optimizing these key verifier performance metrics.

# D TCGBench: Foundational Dataset for TCG Research

As introduced in Section 3.2, **TCGBench** is the comprehensive dataset curated for our Test Case Generation (TCG) research. It aggregates 1840 recent programming problems sourced from three leading competitive programming platforms: AtCoder (https://atcoder.jp), Codeforces (https://codeforces.com), and Nowcoder (https://www.nowcoder.com). These platforms are recognized for their diverse algorithmic challenges. For each problem in TCGBench, we also collected an average of 36.66 incorrect human submissions, specifically those resulting in "Wrong Answer" (WA) or "Time Limit Exceeded" (TLE) verdicts. This large-scale collection of problems, along with their corresponding WA/TLE submissions, provides a rich empirical foundation for studying human error patterns and rigorously developing and evaluating TCG methodologies. The problems are sourced from recent contests to ensure currency and minimize data leakage risks when evaluating contemporary LLMs.

# E TCGBench-Lite and CodeCompass: Curated Set for Evaluation

For the main experimental comparisons and ablation studies presented in Section 4.2.1, and for constructing the **CodeCompass** verifiers (Section 5), we curated **TCGBench-Lite**. This is a focused subset of 270 problems sourced from AtCoder, Codeforces, and Nowcoder contests held since June 2024, ensuring high contemporary relevance and minimizing potential data leakage for evaluating newer models. TCGBench-Lite features an average of 41.41 incorrect submissions per problem.

The difficulty distribution for TCGBench-Lite and CodeCompass (Easy: 27.04%, Medium: 32.59%, Hard: 40.37%) was determined by a multi-faceted approach. This involved considering platform-provided difficulty tags, the type of contest round (e.g., AtCoder Beginner Contest vs. Regular Contest; Codeforces Div.4/3 vs. Div.2/1), typical problem-solving patterns associated with specific problem slots within these contests, and general community perception of difficulty for similar problems. For instance, early problems in beginner-focused contests were generally classified as 'Easy', while later problems in advanced contests or those requiring complex algorithms/data structures were classified as 'Hard'. This classification aims to provide a balanced yet challenging set for rigorous evaluation. The verifiers in CodeCompass, used for code generation evaluation, consist of an average of 50.54 SAGA-generated test cases per problem for these 270 problems. The characteristics are summarized in Table 4.

Table 4: Overview of TCGBench-Lite (Core Dataset for CodeCompass Verifiers).

| Aspect | Details for CodeCompass Evaluation |
|---|---|
| **Core Dataset Source** | Atcoder, Codeforces, Nowcoder (June 2024 - Present) |
| **Total Problems** | 270 |
| **Difficulty Distribution** | Easy (27.04%), Medium (32.59%), Hard (40.37%) |
| **Scale for CG Evaluation** | Avg. Test Cases/Problem: 50.54 |
| | Avg. $\mathcal{S}_{\text{wrong}}$/Problem (for SAGA's TCG): 41.41 |
| **Primary CG Metric** | Pass@k |

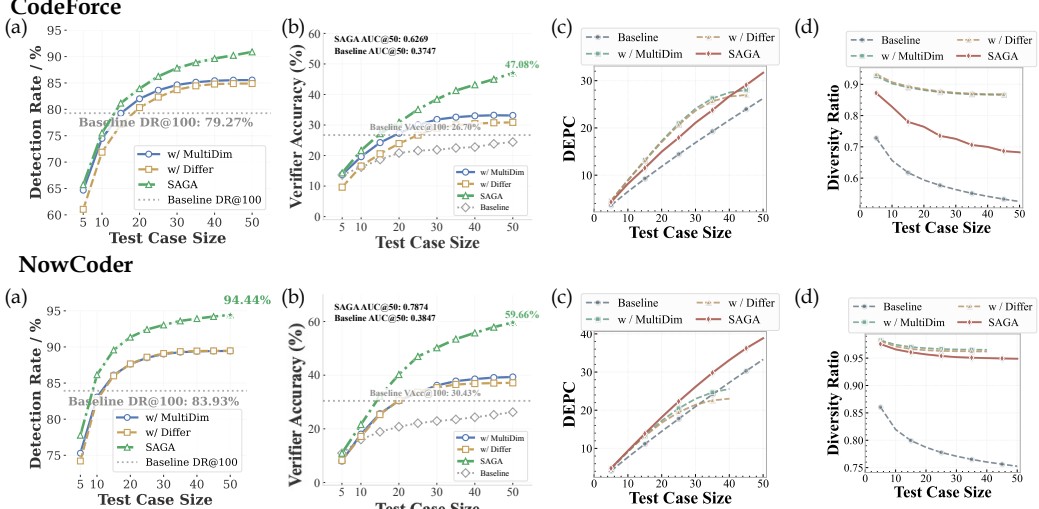

Figure 9: SAGA performance on Codeforces (CF) and Nowcoder (NC) problems from the full TCGBench dataset: (a) Detection Rate, (b) Verifier Accuracy (with AUC@50), (c) DEPC, and (d) Diversity Ratio, compared to Baseline (Input-Interpreter) and SAGA's analytical components (Multidimensional Analysis and Differential Analysis). Dotted lines in (a) & (b) show respective baseline performance at $n = 100$.

# F    SAGA Performance on Full TCGBench

To demonstrate SAGA's broader applicability beyond the AtCoder subset shown in Figure 6 (which used the full TCGBench for that visualization), Figure 9 presents SAGA's performance on the Codeforces and Nowcoder portions of the complete TCGBench dataset. SAGA consistently replicates its superior performance, exhibiting enhanced efficacy (DR, Acc) and superior test suite quality (DEPC, Diversity Ratio) compared to the baseline and its individual analytical components across these platforms as well. This consistent pattern of improvement underscores the fundamental benefits of SAGA's structured, insight-driven approach to TCG.

To further demonstrate SAGA's broad applicability and robustness, Figure 10 presents its Detection Rate (DR) and Verifier Accuracy (VAcc) when paired with different LLM backbones (Qwen2.5-Coder-7B-Instruct, Qwen2.5-72B-Instruct, and DeepSeek-V3-0324) on the Codeforces and Nowcoder portions of the full TCGBench dataset, compared against the Input-Interpreter baseline (using DeepSeek-V3). Across both platforms and all LLM backbones, SAGA consistently and significantly outperforms the baseline in DR and VAcc at various test case sizes. Notably, even SAGA with the smaller Qwen2.5-Coder-7B often surpasses the baseline that utilizes the larger DeepSeek-V3, highlighting SAGA's ability to effectively guide diverse LLMs. While larger SAGA backbones generally yield higher absolute performance, the consistent uplift provided by the SAGA framework across different models and problem sources underscores its fundamental benefits and general applicability for advanced TCG.

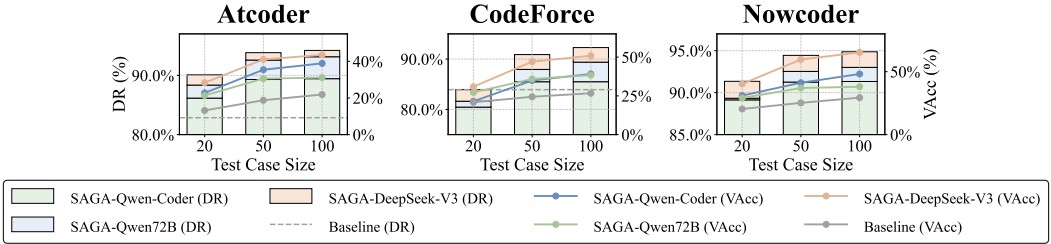

Figure 10: SAGA performance with different LLM backbones (Qwen-Coder, Qwen-72B, DeepSeek-V3) compared to the Baseline (Input-Interpreter with DeepSeek-V3) on Codeforces and Nowcoder portions of the full TCGBench dataset. Metrics: Detection Rate (DR) and Verifier Accuracy (VAcc) at varying test case sizes. Dashed lines indicate baseline performance.

# G  Detailed Performance Analysis on TCGBench-Lite by Difficulty

To provide a more granular understanding of how different Test Case Generation (TCG) methods perform across varying levels of problem complexity, Figure 11 illustrates the Verifier Accuracy (VAcc@50) and Detection Rate (DR@50) of SAGA, its analytical components (Multidimensional Analysis only, denoted "w/ MultiDim"; Differential Analysis only, denoted "w/ Differ"), and baseline TCG methods (TestChain, EvalPlus, and Random Input-Interpreter) on the Easy, Medium, and Hard problem subsets within TCGBench-Lite.

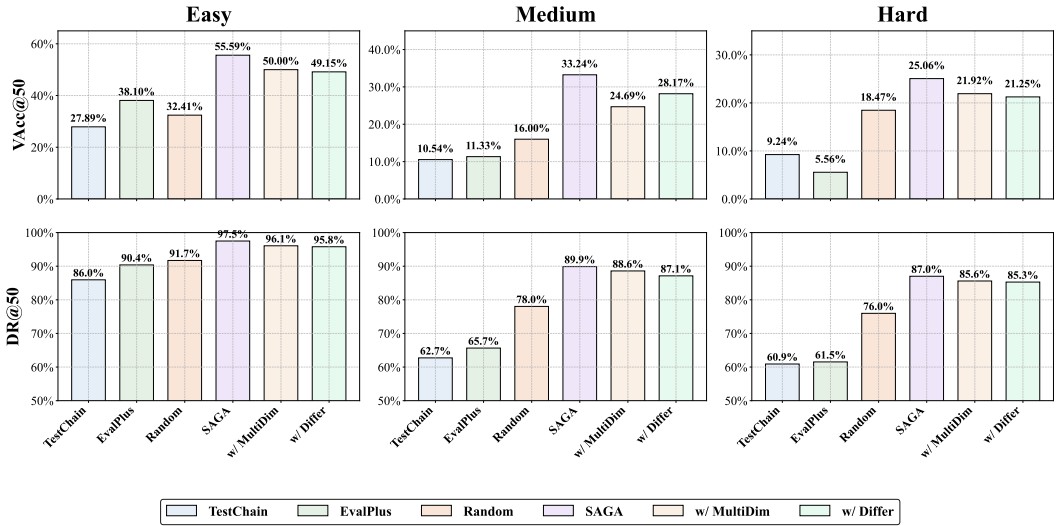

Figure 11: Performance comparison (VAcc@50 and DR@50) of SAGA, its components, and baseline TCG methods across Easy, Medium, and Hard problem subsets of TCGBench-Lite. SAGA consistently outperforms baselines across all difficulties, and its full framework generally surpasses its individual analytical components, especially on harder problems.

**Key Observations from Figure 11:**

- **Consistent Superiority of SAGA:** Across all difficulty tiers (Easy, Medium, Hard) and for both VAcc@50 and DR@50, the full SAGA framework consistently outperforms all baseline methods (TestChain, EvalPlus, Random). This underscores SAGA's robustness and its ability to generate more effective test suites regardless of problem complexity. For instance, on Hard problems, SAGA achieves a VAcc@50 of 25.06%, substantially higher than the Random baseline (18.47%) and other methods. A similar trend is observed for DR@50, where SAGA reaches 87.0% on Hard problems.
- **Impact of Problem Difficulty on Baselines:** The performance of baseline methods, particularly TestChain and EvalPlus, degrades more noticeably as problem difficulty increases. For example, EvalPlus's VAcc@50 drops from 38.10% on Easy problems to a mere 5.56% on Hard problems. This suggests that these methods may struggle to generate effective test cases for more complex scenarios or subtle bugs prevalent in harder problems. The Random Input-Interpreter shows more resilience than TestChain and EvalPlus on harder problems but still falls short of SAGA.
- **Synergy of SAGA's Analytical Components:** While both Multidimensional Analysis ("w/ MultiDim") and Differential Analysis ("w/ Differ") components of SAGA individually outperform the baselines, the full SAGA framework generally achieves the best performance or is highly competitive.
  - On *Easy* problems, SAGA's VAcc@50 (55.59%) is notably higher than "w/ MultiDim" (50.00%) and "w/ Differ" (49.15%), indicating that the combination of insights from both correct and incorrect solutions is beneficial even for simpler problems.
  - For *Medium* problems, SAGA (VAcc@50: 33.24%) again leads, showing a clear advantage over relying on only one type of human prior.
  - On *Hard* problems, the synergy is particularly evident. SAGA's VAcc@50 (25.06%) is superior to "w/ MultiDim" (21.92%) and "w/ Differ" (21.25%). This suggests that for complex problems with elusive bugs, leveraging diverse insights from both correct solution structures and patterns of common errors is crucial for generating highly discriminative test suites.

  A similar synergistic effect is generally observed for DR@50, where the full SAGA framework often provides the highest or near-highest detection rates.
- **Effectiveness of Differential Analysis on Harder Problems:** Interestingly, the "w/ Differ" component (Differential Analysis leveraging $S_{\text{wrong}}$) shows relatively strong performance on Hard problems compared to its performance on Easy problems, particularly for VAcc@50. This might imply that analyzing patterns from incorrect submissions is especially valuable for uncovering the types of subtle or complex errors that characterize more difficult problems.
- **Limitations of Simpler Priors:** The performance of "Random" (Input-Interpreter) and even "EvalPlus" (which uses human solutions for mutation) on Medium and Hard problems highlights that merely having access to human solutions or employing random generation is insufficient. SAGA's structured approach to *analyzing and strategically leveraging* these human priors is what drives its superior performance, especially as complexity increases.

## H   Supplementary Experimental Analyses

To further explore mechanisms for robust test suite construction and the value of diverse generation strategies, we present two additional studies: an analysis of mixing random test cases from different LLMs, and an ablation study on SAGA's knowledge sources.

### H.1   Efficacy of Mixing Random Test Cases from Different Language Models

Our theoretical framework highlights that test suite quality is influenced by individual test potency ($\bar{p}$) and inter-test case correlation ($\bar{\rho}_{\text{eff}}$). We investigated managing $\bar{\rho}_{\text{eff}}$ by mixing random test cases (akin to the Input-Interpreter paradigm from Section 3.2) sourced from different LLMs (V3: DeepSeek-V3-0324, 72B: Qwen2.5-72B-Instruct, Coder: Qwen2.5-Coder-7B-Instruct). The hypothesis is that tests from *different* models may exhibit lower inter-source correlation, leading to improved combined suite characteristics. Figure 12 shows the AUC@50 when mixing random tests, with diagonal elements representing single-LLM suites.

**Observations**

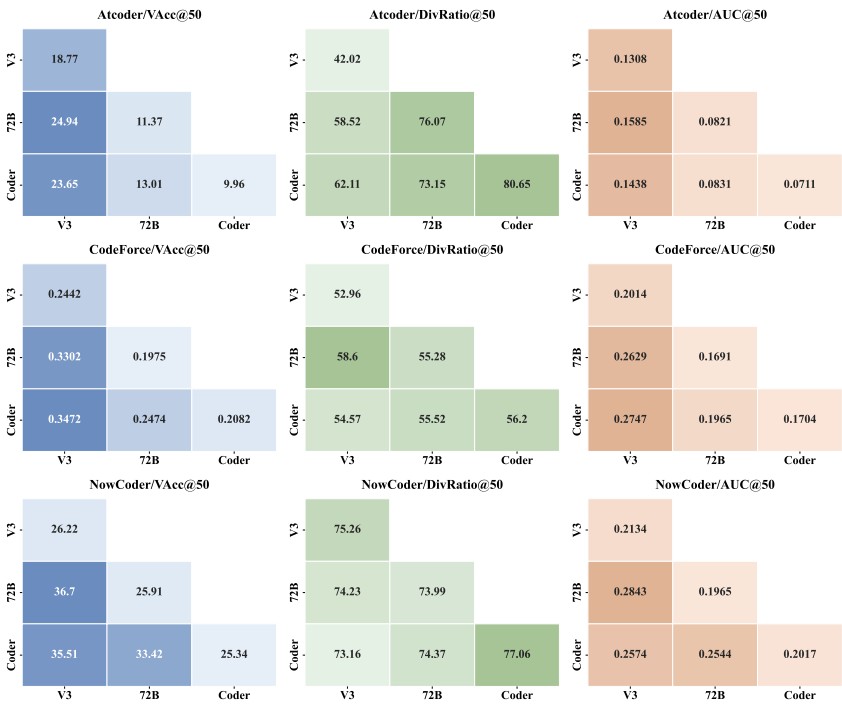

Figure 12: Heatmap illustrating AUC@50 performance of mixed random test suites. Diagonal elements: suites from a single model (V3, 72B, Coder). Off-diagonal (i,j): mixing random tests from model i and model j.

- **Benefit of Model-Source Diversity**: Mixing random tests from two different LLMs frequently yields superior AUC@50 compared to using tests from only one, across AtCoder, Codeforces, and Nowcoder datasets. For instance, on Codeforces, mixing V3 (AUC@50: 0.2014) with 72B (0.1691) results in a mixed AUC@50 of 0.2629, surpassing both. This suggests complementary biases even in random generation.
- **Reduced Effective Correlation**: The improvements imply that combining tests from different models likely lowers $\bar{\rho}_{\text{eff}}$ compared to single-source suites. Enhanced diversity allows the combined suite to cover a broader error spectrum, increasing AUC@50.
- **Surpassing Stronger Components**: Often, the mixed suite outperforms even the stronger individual model in the pair (e.g., V3+72B on Codeforces). This robustly shows different LLMs contribute unique, valuable random tests, highlighting complementary strengths.

**Take-away**: Diversifying the source of even "naive" random tests reduces correlation and improves test suite quality, supporting our theoretical framework. While SAGA achieves this more directly via structured analysis, this experiment underscores the general impact of minimizing $\bar{\rho}_{\text{eff}}$.

### H.2 Ablation Study: Impact of Human Knowledge Source Volume in SAGA

To dissect SAGA's knowledge source contributions, we conducted an ablation study on the Codeforces portion of TCGBench. We modified SAGA by removing its Differential Analysis component (insights from incorrect solutions, $\mathcal{S}_{\text{wrong}}$) entirely. Instead, we doubled the volume of correct human solutions ($\mathcal{S}_{\text{human}}$) fed into SAGA's Multidimensional Analysis component, creating a "Multidim-Enhanced" version. The aim was to see if increasing one type of human insight could compensate for omitting another. Table 5 compares the original SAGA with this "Multidim-Enhanced" configuration.

**Analysis of Ablation Results (Table 5):**

- **Degradation in Core Effectiveness**: Despite doubling $\mathcal{S}_{\text{human}}$ input, the "Multidim-Enhanced" version shows a clear drop in VAcc@50 (from 47.08% to 38.73%) and AUC@50 (from 0.3195 to 0.2744) compared to the full SAGA. This suggests insights from incorrect solutions via Differential Analysis are crucial for verifier quality and cannot be fully compensated by merely increasing the volume of correct solutions for Multidimensional Analysis.

Table 5: Ablation Study on Codeforces (TCGBench): SAGA vs. Multidim-Enhanced (Double $\mathcal{S}_{\text{human}}$ for Multidimensional Analysis, No Differential Analysis).

| Configuration | DR@50 | VAcc@50 | AUC@50 | DivRatio@50 |
|---|---|---|---|---|
| SAGA (Multidim. + Differ.) | 90.89% | 47.08% | 0.3195 | 0.697 |
| Multidim-Enhanced | 88.09% | 38.73% | 0.2744 | 0.701 |

- **Diversity vs. Effectiveness**: While the Diversity Ratio is comparable or slightly higher for "Multidim-Enhanced", this raw diversity doesn't translate to better overall verifier quality (AUC@50). This reinforces that the *type* of diversity and the nature of uncovered error patterns (targeted by Differential Analysis) are critical, not just diversity as a raw count.
- **Implications for SAGA's Design**: This strongly supports SAGA's dual-pronged approach of integrating insights from *both* correct and incorrect human solutions. The unique error patterns revealed by Differential Analysis appear vital for building high-quality, discriminative verifiers, and their contribution is not simply replicable by scaling up the input to Multidimensional Analysis alone.

# I   Model Performance on CodeCompass

To further illustrate the utility of CodeCompass as a challenging benchmark for evaluating LLM code generation capabilities, Table 6 summarizes the Pass@1 performance of several contemporary LLMs. The evaluation was conducted separately for C++ and Python problem instances within CodeCompass, utilizing its SAGA-enhanced verifier suite.

The results demonstrate a clear differentiation among models in both languages, underscoring CodeCompass's ability to effectively rank and assess current code generation models. The challenging nature of the SAGA-generated test cases in CodeCompass provides a rigorous testbed for future model development and comparison.

Table 6: Performance of Various LLMs on CodeCompass (Pass@1) for C++ and Python.

| Model | Pass@1 on CodeCompass (C++) | Pass@1 on CodeCompass (Python) |
|---|---|---|
| Qwen2.5-Coder-32B-Instruct | 15.93% | 12.59% |
| Qwen2.5-72B-Instruct | 17.04% | 14.81% |
| GPT-4o (2024-11-20) | 20.74% | 14.44% |
| DeepSeek-V3 | 24.81% | 23.33% |
| QWQ-32B | 31.85% | 26.30% |
| DeepSeek-Chat-R1 | 38.15% | 34.07% |
| Qwen3-235B-A22B | 43.70% | 36.30% |

From Table 6, it is evident that CodeCompass effectively differentiates LLM performance across both C++ and Python. While relative model rankings show some consistency, language-specific performance variations are also notable, highlighting the benchmark's capacity to reveal nuanced capabilities. The generally moderate Pass@1 rates underscore the challenging nature of CodeCompass due to its SAGA-enhanced test suites, making it a valuable resource for rigorous multi-lingual code generation assessment.

# J   TCGCoder-7B Training Details

TCGCoder-7B, our specialist 7-billion-parameter model for Test Case Generation (TCG), was fine-tuned from **Qwen2.5-Coder-7B-Instruct** [46, 18]. The training dataset, distinct from our evaluation sets to prevent data leakage, comprised 15,000 early-stage programming problems from Codeforces and NowCoder, processed by our SAGA framework (Section 4) to generate structured outputs (Python *Case Scripts*, *Math Explanations*, *Self-Validation* code, per Figure 5). The fine-tuning aimed to distill SAGA's TCG reasoning into TCGCoder-7B. Key training configurations included 3 epochs, a global batch size of 16, an initial learning rate of 5e-6 (minimum 3e-7), a max sequence length of 61,335

tokens, and the qwen2 chat template. Training utilized Fully Sharded Data Parallel (FSDP) across 2 nodes, each with 8 GPUs.

# K  Case Study: SAGA Framework and Metrics in Action

To provide a concrete illustration of our methodology, this section presents a case study demonstrating both the SAGA framework in action and the precise calculation of our proposed evaluation metrics.

## K.1  Illustrative Example of the SAGA Framework

We demonstrate SAGA's dual-pronged analytical approach using the "Christmas Trees" competitive programming problem. The task is to count the number of trees planted at intervals of $M$ starting from coordinate $A$, that fall within the range $[L, R]$.

SAGA systematically leverages both correct and incorrect human solutions to generate a comprehensive verifier.

**1. Multidimensional Analysis (Leveraging Correct Solutions, $\mathcal{S}_{\textbf{human}}$)**  First, by examining a correct human solution, the LLM deconstructs its defensive logic. It identifies the core mathematical condition ($A + k \cdot M \in [L, R]$), analyzes explicit parameter ranges (e.g., $-10^{18} \leq A \leq 10^{18}$), and transforms implicit strategies into formal testing directives. This includes defining boundary values (e.g., testing when $L = A + k \cdot M$) and stress strategies. Guided by these directives, SAGA generates targeted Python *Case Scripts*, such as `gen_TC2()` to test extreme boundary values and `gen_TC3()` for the $L = R$ edge case.

**2. Differential Analysis (Leveraging Incorrect Solutions, $\mathcal{S}_{\textbf{wrong}}$)**  Complementing this, SAGA performs Differential Analysis by comparing a correct solution with a known incorrect one to pinpoint specific failure modes. For the same problem, the LLM identifies a `critical_gap` corresponding to an "Off-by-one error in boundary calculation." It deduces that this failure occurs specifically when the lower bound $L$ is exactly on a tree's position. To exploit this, SAGA generates a targeted test case, `gen_TC1("0 5 10 20")`, where the incorrect solution would fail by one count. Other generated scripts target different identified gaps, such as handling negative ranges or potential overflows with extreme values.

**3. Synthesis and Verification**  SAGA systematically applies both analytical dimensions across a large number of human-provided solutions (both correct and incorrect) for each problem. This generates a comprehensive set of Python *Case Scripts*. These scripts are then executed to produce a large volume of test inputs, which undergo self-validation to ensure adherence to problem constraints. After deduplication, the resulting test suite is verified against a ground-truth human solution to form the final, robust verifier. This structured process ensures the test suite is diverse, targeted, and systematically covers critical logic and subtle failure modes far more effectively than random generation.

## K.2  Illustrative Example of Metric Calculation

To clarify the computation of our core metrics, we use a simple, reproducible example. Consider a problem **P1** for which we have $N_p = 3$ known incorrect solutions, each with a specific bug:

- **S1:** Fails on negative numbers.
- **S2:** Fails due to integer overflow.
- **S3:** Also fails on negative numbers.

Now, suppose a TCG method generates a test suite $T$ with 7 test cases, producing the following outcomes:

**1. Error Pattern Vector, DEPC, and Diversity Ratio:** The **error pattern vector** $v(t_k)$ for each test case is a binary vector of dimension $N_p = 3$, where the $j$-th element is 1 if the test case causes the $j$-th incorrect solution to fail.

**DEPC (Distinct Error Pattern Coverage)** is the number of unique, non-zero error pattern vectors. Here, the set of unique non-zero vectors is $\{`[1, 0, 1]`, `[0, 1, 0]`\}$. Therefore, $\text{DEPC}(T) = 2$.

Table 7: Test case outcomes for problem P1.

| Test Case | Input | Fails S1? | Fails S2? | Fails S3? |
|-----------|-------|-----------|-----------|-----------|
| t1 | -5 | Yes | No | Yes |
| t2 | 999999999 | No | Yes | No |
| t3 | -10 | Yes | No | Yes |
| t4 | 0 | No | No | No |
| t5 | -1 | Yes | No | Yes |
| t6 | 1 | No | No | No |
| t7 | -20 | Yes | No | Yes |

Table 8: Error Pattern Vectors for the test suite $T$.

| Test Case | Error Pattern Vector $v(t_k)$ | Test Case | Error Pattern Vector $v(t_k)$ |
|-----------|-------------------------------|-----------|-------------------------------|
| t1 | '[1, 0, 1]' | t5 | '[1, 0, 1]' |
| t2 | '[0, 1, 0]' | t6 | '[0, 0, 0]' |
| t3 | '[1, 0, 1]' | t7 | '[1, 0, 1]' |
| t4 | '[0, 0, 0]' | | |

The **Diversity Ratio** is $\mathrm{DEPC}(T)/n$, where $n$ is the total number of test cases. For this suite, the Diversity Ratio is $2/7 \approx 0.286$. This indicates that, on average, it takes 3.5 test cases to discover one unique bug pattern, highlighting redundancy in tests t3, t5, t7 and the ineffectiveness of t4, t6.

**2. Detection Rate (DR) and Verifier Accuracy (VAcc): Detection Rate (DR)** is a *solution-level* metric. For each incorrect solution, we determine if the test suite finds the bug (a binary outcome).

- For **S1**: Does $T$ detect the bug? **Yes** (e.g., t1 works). Outcome = 1.
- For **S2**: Does $T$ detect the bug? **Yes** (t2 works). Outcome = 1.
- For **S3**: Does $T$ detect the bug? **Yes** (e.g., t1 works). Outcome = 1.

The final reported DR is the *average* of these outcomes: $\mathrm{DR} = (1 + 1 + 1)/3 = 1.0$ or **100%**.

**Verifier Accuracy (VAcc)** is a stricter, *problem-level* metric. It is 1 if and only if the test suite can detect a failure in *every* known incorrect solution (i.e., if the average DR is 100%). In this example, since the average DR is 100%, $\mathrm{VAcc}(T) = \mathbf{1}$.

## L   Observation: Analysis with a State-of-the-Art Reasoning Model

To explore the impact of advanced reasoning capabilities, we integrated SAGA with a powerful reasoning model, Qwen3-235B-A22B. Counter-intuitively, this led to a degradation in overall performance. A subsequent manual analysis revealed this was not due to a failure in reasoning, but rather a practical engineering constraint. For many complex problems, the model's extensive chain-of-thought process exceeded our deployment's maximum sequence length (64k tokens), resulting in truncated and unusable script outputs. This observation suggests that while powerful, such large-scale models require significant computational resources to fully leverage their reasoning potential in complex generation tasks. Our choice of a smaller default backbone thus reflects a pragmatic balance between performance and accessibility for reproducible research.

## M   Limitations and Future Work

Our primary limitation is SAGA's current reliance on a "golden" ground-truth solution as an interpreter. While effective in domains like competitive programming where oracles are available, this dependency restricts its direct application in general software engineering. Echoing insightful reviewer feedback, a key direction for future work is to adapt SAGA for oracle-less settings. We propose exploring a pseudo-labeling approach where an initial set of diverse test cases—whose quality can be assessed using metrics like DEPC introduced herein—act as "voters" to identify high-confidence "pseudo-gold" solutions from a pool of candidates. These can then seed the SAGA framework, removing the need for a pre-verified oracle and significantly broadening its applicability.

# NeurIPS Paper Checklist

1. **Claims**

   Question: Do the main claims made in the abstract and introduction accurately reflect the paper's contributions and scope?

   Answer: [Yes]

   Justification: The abstract and introduction accurately state the paper's contributions: identifying limitations in current TCG, proposing the SAGA framework, and developing TCG-Bench, CodeComPass, and TCGCoder-7B. These claims are supported by theoretical analysis and extensive experimental results presented throughout the paper.

   Guidelines:
   - The answer NA means that the abstract and introduction do not include the claims made in the paper.
   - The abstract and/or introduction should clearly state the claims made, including the contributions made in the paper and important assumptions and limitations. A No or NA answer to this question will not be perceived well by the reviewers.
   - The claims made should match theoretical and experimental results, and reflect how much the results can be expected to generalize to other settings.
   - It is fine to include aspirational goals as motivation as long as it is clear that these goals are not attained by the paper.

2. **Limitations**

   Question: Does the paper discuss the limitations of the work performed by the authors?

   Answer: [Yes]

   Justification: A dedicated "Limitations and Future Work" section is provided in Appendix **??**, discussing aspects such as the scope of human priors, generalization of TCGCoder-7B, computational cost of SAGA, the dynamic nature of LLMs, subjectivity in difficulty classification, and the current focus on functional correctness.

   Guidelines:
   - The answer NA means that the paper has no limitation while the answer No means that the paper has limitations, but those are not discussed in the paper.
   - The authors are encouraged to create a separate "Limitations" section in their paper.
   - The paper should point out any strong assumptions and how robust the results are to violations of these assumptions (e.g., independence assumptions, noiseless settings, model well-specification, asymptotic approximations only holding locally). The authors should reflect on how these assumptions might be violated in practice and what the implications would be.
   - The authors should reflect on the scope of the claims made, e.g., if the approach was only tested on a few datasets or with a few runs. In general, empirical results often depend on implicit assumptions, which should be articulated.
   - The authors should reflect on the factors that influence the performance of the approach. For example, a facial recognition algorithm may perform poorly when image resolution is low or images are taken in low lighting. Or a speech-to-text system might not be used reliably to provide closed captions for online lectures because it fails to handle technical jargon.
   - The authors should discuss the computational efficiency of the proposed algorithms and how they scale with dataset size.
   - If applicable, the authors should discuss possible limitations of
   - While the authors might fear that complete honesty about limitations might be used by reviewers as grounds for rejection, a worse outcome might be that reviewers discover limitations that aren't acknowledged in the paper. The authors should use their best judgment and recognize that individual actions in favor of transparency play an important role in developing norms that preserve the integrity of the community. Reviewers will be specifically instructed to not penalize honesty concerning limitations.

3. **Theory assumptions and proofs**

Question: For each theoretical result, does the paper provide the full set of assumptions and a complete (and correct) proof?

Answer: [Yes]

Justification: The theoretical analysis concerning detection rate saturation (Corollary 1 in Section 3.2) is presented with its underlying assumptions. The complete derivation, including the model for correlated Bernoulli trials and the proof for the associated theorem, is provided in Appendix C. All theorems, corollaries, and definitions are numbered and cross-referenced.

Guidelines:

- The answer NA means that the paper does not include theoretical results.
- All the theorems, formulas, and proofs in the paper should be numbered and cross-referenced.
- All assumptions should be clearly stated or referenced in the statement of any theorems.
- The proofs can either appear in the main paper or the supplemental material, but if they appear in the supplemental material, the authors are encouraged to provide a short proof sketch to provide intuition.
- Inversely, any informal proof provided in the core of the paper should be complemented by formal proofs provided in appendix or supplemental material.
- Theorems and Lemmas that the proof relies upon should be properly referenced.

4. **Experimental result reproducibility**

Question: Does the paper fully disclose all the information needed to reproduce the main experimental results of the paper to the extent that it affects the main claims and/or conclusions of the paper (regardless of whether the code and data are provided or not)?

Answer: [Yes]

Justification: The paper provides detailed descriptions of the SAGA framework (Section 4.1), the datasets TCGBench and TCGBench-Lite (Appendix E), the CodeComPass verifiers (Section 5), evaluation metrics (Section 3 and Appendix B), and TCGCoder-7B training (Appendix J). The LLM backbones used are specified. Code and data will be released post-review (see Q5).

Guidelines:

- The answer NA means that the paper does not include experiments.
- If the paper includes experiments, a No answer to this question will not be perceived well by the reviewers: Making the paper reproducible is important, regardless of whether the code and data are provided or not.
- If the contribution is a dataset and/or model, the authors should describe the steps taken to make their results reproducible or verifiable.
- Depending on the contribution, reproducibility can be accomplished in various ways. For example, if the contribution is a novel architecture, describing the architecture fully might suffice, or if the contribution is a specific model and empirical evaluation, it may be necessary to either make it possible for others to replicate the model with the same dataset, or provide access to the model. In general. releasing code and data is often one good way to accomplish this, but reproducibility can also be provided via detailed instructions for how to replicate the results, access to a hosted model (e.g., in the case of a large language model), releasing of a model checkpoint, or other means that are appropriate to the research performed.
- While NeurIPS does not require releasing code, the conference does require all submissions to provide some reasonable avenue for reproducibility, which may depend on the nature of the contribution. For example
    (a) If the contribution is primarily a new algorithm, the paper should make it clear how to reproduce that algorithm.
    (b) If the contribution is primarily a new model architecture, the paper should describe the architecture clearly and fully.

    (c) If the contribution is a new model (e.g., a large language model), then there should either be a way to access this model for reproducing the results or a way to reproduce the model (e.g., with an open-source dataset or instructions for how to construct the dataset).

    (d) We recognize that reproducibility may be tricky in some cases, in which case authors are welcome to describe the particular way they provide for reproducibility. In the case of closed-source models, it may be that access to the model is limited in some way (e.g., to registered users), but it should be possible for other researchers to have some path to reproducing or verifying the results.

5. **Open access to data and code**

Question: Does the paper provide open access to the data and code, with sufficient instructions to faithfully reproduce the main experimental results, as described in supplemental material?

Answer: [Yes]

Justification: We commit to releasing the SAGA implementation code, the curated TCG-Bench and TCGBench-Lite datasets, the CodeComPass verifiers, and the TCGCoder-7B model weights upon acceptance of the paper, after the blind review period. All released assets will be accompanied by detailed instructions for use and will be licensed for academic research purposes, with appropriate measures to protect any residual privacy concerns from the original public data sources. Anonymous repository links for other code, prompt templates, and the data demo are now on the first page.

Guidelines:

- The answer NA means that paper does not include experiments requiring code.
- Please see the NeurIPS code and data submission guidelines (`https://nips.cc/public/guides/CodeSubmissionPolicy`) for more details.
- While we encourage the release of code and data, we understand that this might not be possible, so "No" is an acceptable answer. Papers cannot be rejected simply for not including code, unless this is central to the contribution (e.g., for a new open-source benchmark).
- The instructions should contain the exact command and environment needed to run to reproduce the results. See the NeurIPS code and data submission guidelines (`https://nips.cc/public/guides/CodeSubmissionPolicy`) for more details.
- The authors should provide instructions on data access and preparation, including how to access the raw data, preprocessed data, intermediate data, and generated data, etc.
- The authors should provide scripts to reproduce all experimental results for the new proposed method and baselines. If only a subset of experiments are reproducible, they should state which ones are omitted from the script and why.
- At submission time, to preserve anonymity, the authors should release anonymized versions (if applicable).
- Providing as much information as possible in supplemental material (appended to the paper) is recommended, but including URLs to data and code is permitted.

6. **Experimental setting/details**

Question: Does the paper specify all the training and test details (e.g., data splits, hyperparameters, how they were chosen, type of optimizer, etc.) necessary to understand the results?

Answer: [Yes]

Justification: The paper provides details on dataset construction (TCGBench, TCGBench-Lite, CodeComPass in Appendix E, D and Section 5), LLMs used for experiments (Section 3.2, Section 4.2.1), and key training parameters for TCGCoder-7B (Appendix J). The experimental setup for SAGA and baselines is described, enabling an understanding of the results.

Guidelines:

- The answer NA means that the paper does not include experiments.

- The experimental setting should be presented in the core of the paper to a level of detail that is necessary to appreciate the results and make sense of them.
- The full details can be provided either with the code, in appendix, or as supplemental material.

7. **Experiment statistical significance**

Question: Does the paper report error bars suitably and correctly defined or other appropriate information about the statistical significance of the experiments?

Answer: [Yes]

Justification: Error regions are shown in Figure 4(b) for the experimental validation of the theoretical upper bound on detection rate, illustrating the consistency of the observed trend. For the main experimental results (Acc, DR) presented in Table 2, error bars representing standard deviation across multiple runs or problem subsets will be included in the Appendix **??** to provide information on result stability.

Guidelines:

- The answer NA means that the paper does not include experiments.
- The authors should answer "Yes" if the results are accompanied by error bars, confidence intervals, or statistical significance tests, at least for the experiments that support the main claims of the paper.
- The factors of variability that the error bars are capturing should be clearly stated (for example, train/test split, initialization, random drawing of some parameter, or overall run with given experimental conditions).
- The method for calculating the error bars should be explained (closed form formula, call to a library function, bootstrap, etc.)
- The assumptions made should be given (e.g., Normally distributed errors).
- It should be clear whether the error bar is the standard deviation or the standard error of the mean.
- It is OK to report 1-sigma error bars, but one should state it. The authors should preferably report a 2-sigma error bar than state that they have a 96% CI, if the hypothesis of Normality of errors is not verified.
- For asymmetric distributions, the authors should be careful not to show in tables or figures symmetric error bars that would yield results that are out of range (e.g. negative error rates).
- If error bars are reported in tables or plots, The authors should explain in the text how they were calculated and reference the corresponding figures or tables in the text.

8. **Experiments compute resources**

Question: For each experiment, does the paper provide sufficient information on the computer resources (type of compute workers, memory, time of execution) needed to reproduce the experiments?

Answer: [Yes]

Justification: Appendix J specifies the GPU resources (2 nodes, 8 GPUs each, CUDA 12.2.2 environment) for TCGCoder-7B training. LLM inference for SAGA and baseline experiments was conducted on similar high-end GPU hardware (e.g., NVIDIA A100s). While precise execution times per problem vary, the scale of LLMs used implies standard computational requirements for models of these sizes.

Guidelines:

- The answer NA means that the paper does not include experiments.
- The paper should indicate the type of compute workers CPU or GPU, internal cluster, or cloud provider, including relevant memory and storage.
- The paper should provide the amount of compute required for each of the individual experimental runs as well as estimate the total compute.
- The paper should disclose whether the full research project required more compute than the experiments reported in the paper (e.g., preliminary or failed experiments that didn't make it into the paper).

9. **Code of ethics**

Question: Does the research conducted in the paper conform, in every respect, with the NeurIPS Code of Ethics https://neurips.cc/public/EthicsGuidelines?

Answer: [Yes]

Justification: The research adheres to the NeurIPS Code of Ethics. It focuses on improving evaluation methodologies for AI-generated code. Data is sourced from public programming platforms, and care will be taken in any release to ensure user privacy from these original sources is respected (e.g., by anonymizing any potentially identifying information if present, though the focus is on code logic).

Guidelines:

- The answer NA means that the authors have not reviewed the NeurIPS Code of Ethics.
- If the authors answer No, they should explain the special circumstances that require a deviation from the Code of Ethics.
- The authors should make sure to preserve anonymity (e.g., if there is a special consideration due to laws or regulations in their jurisdiction).

10. **Broader impacts**

Question: Does the paper discuss both potential positive societal impacts and negative societal impacts of the work performed?

Answer: [Yes]

Justification: The paper primarily focuses on the positive societal impact of enabling more reliable and robust AI-assisted software development through improved evaluation (Conclusion, Section 6). Potential negative impacts (e.g., accelerating automation) are briefly acknowledged in Appendix **??**. The work aims to improve AI safety and reliability in coding.

Guidelines:

- The answer NA means that there is no societal impact of the work performed.
- If the authors answer NA or No, they should explain why their work has no societal impact or why the paper does not address societal impact.
- Examples of negative societal impacts include potential malicious or unintended uses (e.g., disinformation, generating fake profiles, surveillance), fairness considerations (e.g., deployment of technologies that could make decisions that unfairly impact specific groups), privacy considerations, and security considerations.
- The conference expects that many papers will be foundational research and not tied to particular applications, let alone deployments. However, if there is a direct path to any negative applications, the authors should point it out. For example, it is legitimate to point out that an improvement in the quality of generative models could be used to generate deepfakes for disinformation. On the other hand, it is not needed to point out that a generic algorithm for optimizing neural networks could enable people to train models that generate Deepfakes faster.
- The authors should consider possible harms that could arise when the technology is being used as intended and functioning correctly, harms that could arise when the technology is being used as intended but gives incorrect results, and harms following from (intentional or unintentional) misuse of the technology.
- If there are negative societal impacts, the authors could also discuss possible mitigation strategies (e.g., gated release of models, providing defenses in addition to attacks, mechanisms for monitoring misuse, mechanisms to monitor how a system learns from feedback over time, improving the efficiency and accessibility of ML).

11. **Safeguards**

Question: Does the paper describe safeguards that have been put in place for responsible release of data or models that have a high risk for misuse (e.g., pretrained language models, image generators, or scraped datasets)?

Answer: [Yes]

Justification: The datasets (TCGBench, TCGBench-Lite, CodeComPass) are derived from publicly available programming problems. Upon release, any data derived from user submissions (e.g., for $\mathcal{S}_{\text{wrong}}$) will be processed to remove or anonymize any potentially identifying information to protect user privacy, ensuring they are used solely for academic research. The TCGCoder-7B model is specialized for TCG and does not pose the same general misuse risks as large foundation models.

Guidelines:

- The answer NA means that the paper poses no such risks.
- Released models that have a high risk for misuse or dual-use should be released with necessary safeguards to allow for controlled use of the model, for example by requiring that users adhere to usage guidelines or restrictions to access the model or implementing safety filters.
- Datasets that have been scraped from the Internet could pose safety risks. The authors should describe how they avoided releasing unsafe images.
- We recognize that providing effective safeguards is challenging, and many papers do not require this, but we encourage authors to take this into account and make a best faith effort.

12. **Licenses for existing assets**

Question: Are the creators or original owners of assets (e.g., code, data, models), used in the paper, properly credited and are the license and terms of use explicitly mentioned and properly respected?

Answer: [Yes]

Justification: The paper properly cites original sources for all baseline methods (e.g., LiveCodeBench [20], EvalPlus [28], TestChain [23]) and LLMs used (e.g., Qwen series [46, 18], DeepSeek models [8, 13]). The programming problems for TCGBench are from public contest platforms (AtCoder, Codeforces, Nowcoder), whose terms generally permit research use. The open-source LLMs used typically have licenses like Apache 2.0, which are respected.

Guidelines:

- The answer NA means that the paper does not use existing assets.
- The authors should cite the original paper that produced the code package or dataset.
- The authors should state which version of the asset is used and, if possible, include a URL.
- The name of the license (e.g., CC-BY 4.0) should be included for each asset.
- For scraped data from a particular source (e.g., website), the copyright and terms of service of that source should be provided.
- If assets are released, the license, copyright information, and terms of use in the package should be provided. For popular datasets, paperswithcode.com/datasets has curated licenses for some datasets. Their licensing guide can help determine the license of a dataset.
- For existing datasets that are re-packaged, both the original license and the license of the derived asset (if it has changed) should be provided.
- If this information is not available online, the authors are encouraged to reach out to the asset's creators.

13. **New assets**

Question: Are new assets introduced in the paper well documented and is the documentation provided alongside the assets?

Answer: [Yes]

Justification: The SAGA framework, TCGBench dataset, TCGBench-Lite subset, CodeComPass verifiers, and TCGCoder-7B model are introduced. Their construction, characteristics, and methodologies are documented (Sections 4.1, 5, Appendices D, J, E). Upon acceptance and release, comprehensive documentation will accompany all assets.

Guidelines:

- The answer NA means that the paper does not release new assets.
- Researchers should communicate the details of the dataset/code/model as part of their submissions via structured templates. This includes details about training, license, limitations, etc.
- The paper should discuss whether and how consent was obtained from people whose asset is used.
- At submission time, remember to anonymize your assets (if applicable). You can either create an anonymized URL or include an anonymized zip file.

14. **Crowdsourcing and research with human subjects**

Question: For crowdsourcing experiments and research with human subjects, does the paper include the full text of instructions given to participants and screenshots, if applicable, as well as details about compensation (if any)?

Answer: [NA]

Justification: This research does not involve crowdsourcing or direct experiments with human subjects. The "human solutions" referred to are publicly available code submissions from programming contest platforms, used in an aggregated and anonymized fashion for analysis.

Guidelines:

- The answer NA means that the paper does not involve crowdsourcing nor research with human subjects.
- Including this information in the supplemental material is fine, but if the main contribution of the paper involves human subjects, then as much detail as possible should be included in the main paper.
- According to the NeurIPS Code of Ethics, workers involved in data collection, curation, or other labor should be paid at least the minimum wage in the country of the data collector.

15. **Institutional review board (IRB) approvals or equivalent for research with human subjects**

Question: Does the paper describe potential risks incurred by study participants, whether such risks were disclosed to the subjects, and whether Institutional Review Board (IRB) approvals (or an equivalent approval/review based on the requirements of your country or institution) were obtained?

Answer: [NA]

Justification: The research does not involve direct human subject participation that would require IRB approval. The data used (code submissions) is from public sources where users have agreed to platform terms; our use is for aggregated analysis and TCG methodology development, with privacy considerations for any release as noted in Q11.

Guidelines:

- The answer NA means that the paper does not involve crowdsourcing nor research with human subjects.
- Depending on the country in which research is conducted, IRB approval (or equivalent) may be required for any human subjects research. If you obtained IRB approval, you should clearly state this in the paper.
- We recognize that the procedures for this may vary significantly between institutions and locations, and we expect authors to adhere to the NeurIPS Code of Ethics and the guidelines for their institution.
- For initial submissions, do not include any information that would break anonymity (if applicable), such as the institution conducting the review.

16. **Declaration of LLM usage**

Question: Does the paper describe the usage of LLMs if it is an important, original, or non-standard component of the core methods in this research? Note that if the LLM is used only for writing, editing, or formatting purposes and does not impact the core methodology, scientific rigorousness, or originality of the research, declaration is not required.

Answer: [Yes]

Justification: LLMs are central to this research. The SAGA framework (Section 4.1) is a human-LLM collaborative method for TCG. LLMs are also used in the baseline TCG methods we evaluate (Section 3.2) and are the subjects of the improved evaluation methodologies proposed. Specific LLMs (e.g., DeepSeek-V3-0324, Qwen series) are mentioned as backbones for these processes.

Guidelines:

- The answer NA means that the core method development in this research does not involve LLMs as any important, original, or non-standard components.
- Please refer to our LLM policy (`https://neurips.cc/Conferences/2025/LLM`) for what should or should not be described.

