# OpenReview forum: "Rethinking Verification for LLM Code Generation: From Generation to Testing"
_NeurIPS.cc/2025/Conference — NeurIPS 2025 poster_

### Official Review · Reviewer_2jtr · 2025-06-27

**Clarity:** 2
**Significance:** 3
**Originality:** 2
**Rating:** 3
**Confidence:** 3

**Summary:**

The paper provides a critique of Test Case Generation (TCG) for measuring LLM-based code generation. The critique asserts that existing benchmarks suffer from homogeneity of test cases that, in some cases, allow subtle faults to go undetected.  The paper defined four (4) metrics for test case quality.  Test case quality was studied by assembling and measuring results from TCGBench, a benchmark aggregated from existing code generation benchmarks. The paper then introduces Strategic Adversarial & Constraint-differential GenerAtive (SAGA), a workflow that combines LLM and human insight to generate improved test suites using novel methods for multi-dimensional and differential analysis. The SAGA method is then used to develop CodeComPass (a verifier) and TCGCoder-7B (a test case generation model).

**Questions:**

No questions.

**Ethical Concerns:**

["NO or VERY MINOR ethics concerns only"]

**Final Justification:**

I appreciated the author's rebuttal and their addressing my feedback. I have also looked at the material provided as response to other reviewer's comments. On the basis of these clarifications and improvements I have increased my overall rating to 3.

**Limitations:**

The lack of a discussion on related work is a critical defect. The paper includes a reasonable section on limitations, but as Appendix J. Limitations are not addressed or summarized in the body of the paper.

**Paper Formatting Concerns:**

No formatting concerns noted.

**Quality:**

3

**Strengths And Weaknesses:**

Strength - The paper starts with a relevant and important question - "Are the test cases of current benchmarks for evaluating models’ code capabilities robust enough?"

Weakness - A significant defect in this work is that there is no discussion of related work in the body of the paper. This includes both work that has been done to measure/improve individual benchmarks like HumanEval and work to measure/improve the quality of code assessment generally.

Weakness - Section 2 is largely a restatement of the limitations of empirical testing (in contrast to formal methods). The section introduces definitions for DR and VAcc, but then concludes by defining two (2) new additional metrics. What constitutes a distinct error pattern is defined, by not well explained in a practical setting.

Weakness - Section 3 begins with the claim "Section 2.2 revealed critical limitations in TGC paradigms, such as low test quality and homogeneity. It is not clear how this statement follows from what was presented in Section 2.2 or Figure 4.

Weakness - While the idea of leveraging human effort (especially already expended human effort) is a good one, and while the method shows improved testing capability, there isn't enough detail to clearly understand (or reproduce) what was done. The descriptions of Multidimensional and Differential Analysis are brief and vague.  Since this is one of the main contributions of the paper, there should be more here.

Strength - It would seem that the paper provides a path toward generating a better verifier approach for existing and to-be-constructed benchmarks that would make for a valuable paper, but the result is a bit muddled in its presentation. Weaknesses as described should be addressed.  It might be that the measurement and improvement (SAGA) portions should be separate papers.

---

> ### Author Rebuttal · Authors · 2025-07-31
>
> Dear Reviewer 2jtr,
>
> We thank you for your feedback, and especially for recognizing the **importance and relevance** of the question we address, as well as the **potential path and value** our work offers for building better verifiers. Your comments have helped us identify several areas where the presentation clarity and impact of our paper can be substantially improved. We assure you that we have taken all your points seriously and have prepared the following clarifications, which we hope will address your concerns. We sincerely thank you for the time and effort you have invested in reviewing our manuscript.
>
> We will address the weaknesses you pointed out one by one:
>
> **Weakness 1: "no discussion of related work"**
>
> Thank you for pointing out this issue. We would like to clarify right away: **we do have a very comprehensive related work section, located in Appendix A, and we commit to moving a condensed summary into the main body in the final version, adopting a "main text summary + appendix details" structure.**
>
> The reason for this initial arrangement was that the main text space was insufficient to accommodate the full version. In order not to sacrifice the depth and completeness of the review—which we feel is critical for readers to fully understand the landscape—we made the difficult decision to place the full discussion in **Appendix A**. This section (A.1 and A.2) covers advancements in LLM-based code generation, various evaluation benchmarks (like HumanEval, EvalPlus), and both traditional (SBST, Fuzzing) and LLM-based TCG methodologies. We sincerely apologize that this information was not easily discoverable. In the final version of the paper, we will move a condensed yet thorough summary of this related work section into the main body to ensure all readers can understand how our work is situated within the broader academic context. Furthermore, we will cite the latest research in the updated version and **sincerely welcome any relevant references you feel we may have overlooked**. We will incorporate them into our final revision.
>
> **Weakness 2: Clarity of Section 2 and its metrics**
>
> We appreciate your feedback on the clarity of Section 2. Our initial intention was to build a comprehensive measurement framework, and we now recognize that our initial explanation was too dense. The four metrics (DR, VAcc, DEPC, AUC-AccN) are designed to assess test suite quality from multiple perspectives.
>
> -   DR and VAcc measure **effectiveness** (Can it find bugs?).
> -   DEPC and AUC-AccN measure **intrinsic quality and efficiency** (How diverse and potent are the tests?).
>     The "error pattern vector" is a practical method for defining diversity: it is a vector that represents which incorrect programs a single test case can detect a bug in. We have provided more detailed descriptions of all these metrics, their formulas, and their interpretations in **Appendix B**. Based on your feedback, we will revise Section 2 in the main text to provide more intuitive explanations and examples, referencing the clearer descriptions in the appendix.
> -   The practical value of these metrics is demonstrated throughout the paper:
>     -   In **Figure 6**, we use them to precisely show ***why*** SAGA is superior to the baselines (as it achieves higher DEPC and AUC-AccN).
>     -   **Appendix B** provides their full mathematical derivations and practical interpretations.
>
> **Weakness 3: The logical link between Section 2.2 and Section 3**
>
> The assertion at the beginning of Section 3 that "Section 2.2 revealed limitations" is directly supported by the experimental results presented in that very section, specifically **Figure 3 and Figure 4**.
>
> -   **Figure 3** shows the *low quality* of directly generated tests (low DR and VAcc).
> -   **Figure 4** shows the *homogeneity* of tests from the Input-Interpreter paradigm, evidenced by detection rate saturation, which validates the theoretical upper bound for the most commonly used paradigm today.
>     These results demonstrate what we refer to as "low test quality and homogeneity." We will rewrite the opening of Section 3 to explicitly reference these figures and findings, thereby establishing a clearer motivation for SAGA for the reader.
>
> **Weakness 4: Lack of detail on the SAGA method**
>
> Thank you for the emphasis. The power of SAGA lies in its structured, human-insight-driven process. The core mechanism of SAGA operates through **detailed, structured prompts, which we have open-sourced in the supplementary materials** (with full prompts for both Multidimensional and Differential Analysis). These prompts are effectively the *"source code"* of our method. To make it completely transparent and reproducible, we will **add a new, detailed running example to the appendix** of the final paper. This example will demonstrate:
>
> 1.  How a problem description, a correct human solution, and a buggy human solution are fed into the SAGA prompts.
> 2.  The LLM's structured output, including the `constraint_analysis` and the generated `Case Scripts`.
> 3.  How these scripts are executed to produce the final test cases.
>     This will provide the concrete, reproducible detail that you rightly pointed out was missing.
>
> **Limitations:**
>
> Thank you for your attention to this. A dedicated and detailed discussion of limitations and future work is located in **Appendix J**. We discuss the scope of human priors, generalization, computational cost, and other important factors. We recognize the importance of highlighting these limitations and will **add a summary of this section to the conclusion in the main body** to ensure this crucial context is not missed by readers.
>
> We are confident that these clarifications and the planned revisions, directly guided by your valuable feedback, will address the issues you have raised and significantly strengthen our paper. Thank you again for your rigorous review and for helping us improve our work.
>
> Best regards,
>
> Authors of Submission 17344

---

> > ### Comment · Reviewer_2jtr · 2025-08-04
> >
> > Thank you for your rebuttal and for addressing all of my feedback. I understand your comments about the related work and limitations sections. Having the bulk of these sections in the appendix is fine, but a summary in the body provides useful context. Thank you for agreeing to the changes. Similarly, I think the paper will be improved by a running example, as other reviewers also noted. Thank you for agreeing to make the addition in the appendix.
> >
> > I appreciate the clarifications on the definitions and use of the metrics and the connection between Section 3 and Figures 3 and 4. Even having reviewed your comments and Appendix B, I still feel there is a lack of specificity and reproducibility in the error pattern vector calculation, which is important to the method. An example here would be helpful.

---

> > > ### Author Response · Authors · 2025-08-05
> > > **An example for our proposed metric.(Part 1)**
> > >
> > > Dear Reviewer 2jtr,
> > >
> > > Thank you for your thoughtful follow-up. We sincerely appreciate your continued engagement.
> > >
> > > You raised the point that the error pattern vector calculation for DEPC needs an example for support. To address this, we have prepared a simple, running example below that illustrates how **all four of our key metrics (DR, VAcc, DEPC, and AUC-AccN)** are calculated, paying special attention to the precise definition of Detection Rate as used in our paper. We will add this detailed explanation to a new section in the appendix.
> > >
> > > ---
> > >
> > > #### **A Concrete and Reproducible Example for Our Metrics Framework**
> > >
> > > Our framework is designed to be fully reproducible. The key is that for any given problem *P*, we have a fixed set of *Np* known incorrect human solutions, $S_{wrong}$ = {$S_1$,$S_2$, ..., $S_{Np}$}. This set forms the basis for our evaluation.
> > >
> > > Let's consider a coding problem **P1**, for which we have collected **3 known incorrect solutions** (*Np*=3). Each solution has a specific bug:
> > >
> > > *   **S1:** Fails on **negative numbers**.
> > > *   **S2:** Fails due to **integer overflow**.
> > > *   **S3:** Also fails on **negative numbers**.
> > >
> > > Now, suppose a method generates a test suite **T** with **7 test cases**: {t1, t2, ..., t7}.
> > >
> > > | Test Case | Input       | Fails S1? | Fails S2? | Fails S3? |
> > > | :-------- | :---------- | :-------: | :-------: | :-------: |
> > > | t1        | `-5`        |  **Yes**  |    No     |  **Yes**  |
> > > | t2        | `999999999` |    No     |  **Yes**  |    No     |
> > > | t3        | `-10`       |  **Yes**  |    No     |  **Yes**  |
> > > | t4        | `0`         |    No     |    No     |    No     |
> > > | t5        | `-1`        |  **Yes**  |    No     |  **Yes**  |
> > > | t6        | `1`         |    No     |    No     |    No     |
> > > | t7        | `-20`       |  **Yes**  |    No     |  **Yes**  |
> > >
> > > ---
> > >
> > > #### **1. Error Pattern Vector, DEPC, and Diversity Ratio (Your Core Concern)**
> > >
> > > The **error pattern vector `v(tk)`** is a binary vector whose **dimension is fixed for a given problem** (here, the dimension is *Np*=3). For each test case `tk`, the j-th element of its vector is 1 if it causes the j-th incorrect solution (`Sj`) to fail, and 0 otherwise. This vector provides a **reproducible signature** of the bug-finding capability of a test case for that specific problem.
> > >
> > > Based on the table above, the error pattern vectors for problem **P1** are:
> > >
> > > | Test Case | Error Pattern Vector `v(tk)` |
> > > | :-------- | :--------------------------- |
> > > | t1        | `[1, 0, 1]`                  |
> > > | t2        | `[0, 1, 0]`                  |
> > > | t3        | `[1, 0, 1]`                  |
> > > | t4        | `[0, 0, 0]`                  |
> > > | t5        | `[1, 0, 1]`                  |
> > > | t6        | `[0, 0, 0]`                  |
> > > | t7        | `[1, 0, 1]`                  |
> > >
> > > **DEPC (Distinct Error Pattern Coverage)** is the number of *unique, non-zero* error pattern vectors.
> > >
> > > *   The set of unique, non-zero vectors is: `{ [1, 0, 1], [0, 1, 0] }`
> > > *   Therefore, for test suite T on problem P1, **DEPC(T) = 2**.
> > >
> > > **Diversity Ratio** is then defined as **DEPC(T) / n**, where *n* is the total number of test cases.
> > >
> > > *   In our example, n = 7.
> > > *   So, the **Diversity Ratio = 2 / 7 ≈ 0.286**.
> > >
> > > ***Reproducibility and Deeper Insight:***
> > > This ratio provides a crucial, normalized measure of a test suite's **efficiency**. A Diversity Ratio of 0.286 means that, on average, **it takes roughly 3.5 test cases (7 tests / 2 unique patterns) to contribute one *new, unique* way of finding bugs.** This clearly shows that tests `t3, t5, t7` are **functionally redundant** with `t1`, and `t4, t6` are ineffective. A "smarter" test suite like `{t1, t2}` would have a perfect Diversity Ratio of 1.0, indicating maximum efficiency. This metric allows us to quantitatively identify such bloated and inefficient test suites.
> > >
> > > If we were evaluating a different problem, **P2**, with *100* known incorrect solutions, the error pattern vector would be **100-dimensional**, but the calculation and interpretation of DEPC and Diversity Ratio would remain identical, ensuring the metric is both reproducible and scalable.

---

> > > > ### Author Response · Authors · 2025-08-05
> > > > **An example for our proposed metric.(Part 2)**
> > > >
> > > > #### **2. Detection Rate (DR) and Verifier Accuracy (VAcc)**
> > > >
> > > > This is the area where your feedback prompted a crucial clarification.
> > > >
> > > > *   **Detection Rate (DR):** As defined in our paper, DR is a **solution-level metric**. This means for *each individual* incorrect solution `Si`, we determine if our test suite `T` can detect its bug. The outcome is binary: **1 if detected, 0 if not.** The final DR score that we report (e.g., in Table 2) is the **average of these binary outcomes across all incorrect solutions**.
> > > >
> > > > Let's calculate it step-by-step for our example:
> > > >
> > > > 1.  **For solution S1:** Does `T` contain at least one test case that makes S1 fail? **Yes** (t1, t3, t5, t7 all work). So, the detection outcome for S1 is **1**.
> > > > 2.  **For solution S2:** Does `T` contain at least one test case that makes S2 fail? **Yes** (t2 works). The detection outcome for S2 is **1**.
> > > > 3.  **For solution S3:** Does `T` contain at least one test case that makes S3 fail? **Yes** (t1, t3, t5, t7 all work). The detection outcome for S3 is **1**.
> > > >
> > > > The **Average Detection Rate (DR)** for this problem is the mean of these outcomes:
> > > > `DR = (1 + 1 + 1) / 3 = 1.0` or **100%**.
> > > >
> > > > *To illustrate further:* If our test suite **T'** did *not* contain test case `t2`, the outcomes would be:
> > > >
> > > > *   S1 detected: 1
> > > > *   S2 detected: **0**
> > > > *   S3 detected: 1
> > > >     The Average DR for suite **T'** would be `(1 + 0 + 1) / 3 = 0.667` or **66.7%**.
> > > >
> > > > *   **Verifier Accuracy (VAcc):** This is a stricter, **problem-level** metric. It is 1 if and only if the test suite T can detect a failure in *every single* incorrect solution. It is essentially asking if the Average DR is 100%.
> > > >     *   In our original example with suite **T**, the Average DR is 100%, so **VAcc(T) = 1**.
> > > >     *   In the counter-example with suite **T'**, the Average DR is 66.7%, so **VAcc(T') = 0**.
> > > >
> > > > ---
> > > >
> > > > #### **3. Normalized Area Under the Curve (AUC-AccN)**
> > > >
> > > > This metric measures the *efficiency* of a test suite by tracking how VAcc changes as tests are added one by one. A test suite that achieves VAcc=1 faster will have a higher **AUC-AccN** score, reflecting a superior combination of test *potency* and *diversity*.

---

> > > > > ### Author Response · Authors · 2025-08-05
> > > > > **An example for our proposed metric.(Part 3)**
> > > > >
> > > > > We hope this detailed, step-by-step example, which explicitly addresses the **reproducibility of the error pattern vector** and the **precise calculation of Average Detection Rate**, fully clarifies the intuition behind our metrics. We have taken your feedback very seriously.
> > > > >
> > > > > We will add this example to Appendix B and include new visualizations to help readers understand the calculation of this metric more intuitively. We are keen to know if all your concerns have been addressed and would be eager for any further feedback you might have. :)
> > > > >
> > > > > Thank you once again for your invaluable guidance.

---

> ### Author Response · Authors · 2025-08-07
>
> Dear Reviewer 2jtr,
>
> Thank you again for your invaluable guidance. Following your suggestion, we have posted a detailed response with a concrete example for our metrics.
>
> We are very keen to ensure all your concerns are fully addressed. Please let us know if any part of our explanation remains unclear, as we would be grateful for the opportunity to provide further clarification.
>
> Best regards

---

> ### Author Response · Authors · 2025-08-08
> **Kindly Reminder**
>
> Thank you for your efforts and reviews. With the discussion now drawing to a close, we would be grateful to know whether our response has fully addressed your concerns.
>
> If there are any further questions or areas you feel need additional clarification, please do not hesitate to let us know. We are more than happy to continue the discussion to address them promptly.
>
> Conversely, if you feel that all your concerns have now been clarified, we would sincerely appreciate it if you would consider updating your evaluation to reflect the progress made.
>
> Thank you again for your time and engagement throughout this process.
>
> Best regards,
>
> Authors of Submission 17344

---

### Official Review · Reviewer_Y2Dv · 2025-07-01

**Clarity:** 3
**Significance:** 3
**Originality:** 3
**Rating:** 5
**Confidence:** 4

**Summary:**

This paper systematically analyzes the limitations of test cases in current LLM code generation evaluation, and proposes a human-LLM collaborative test case generation framework SAGA. SAGA generates high-coverage and high-discrimination test cases by integrating correct/incorrect human programming solutions. The authors construct a TCGBench dataset for evaluation, and develop a new benchmark CodeComPass and a dedicated model TCGCoder-7B based on SAGA. Experiments show that SAGA significantly outperforms existing methods (such as LiveCodeBench) in detection rate (DR) and verifier accuracy (VAcc), and CodeComPass can more rigorously evaluate LLM code generation capabilities.

**Questions:**

Overall, this is a work of good quality. I have some concerns that the author can address and further enhance the paper accordingly.

First, how efficient is SAGA? It seems SAGA requires multiple LLM calls, thus, high computational and monetary cost. Please justify it.

Then, SAGA requires correct/incorrect human codes, which introduces another cost, i.e., human effort to collect such programs. It is easy to collect this kind of program from competition-level datasets such as Codeforces. However, for real-world software projects, it will be challenging. As a result, the generalization ability of SAGA is unclear, which should be discussed.

Again, the evaluations are based solely on competition-level datasets, which hinders the conclusion of the experiments.

TCGBench-Lite selects datasets after June 2024 to avoid data leakage, but does not explain how to isolate the evaluation set when training TCGCoder-7B. Please clarify the training/validation/test set division of TCGCoder-7B.

**Ethical Concerns:**

["NO or VERY MINOR ethics concerns only"]

**Final Justification:**

Good work. Please consider adding the discussions in the response in the final version.

**Limitations:**

yes

**Quality:**

3

**Strengths And Weaknesses:**

Strength

+ Target an important problem – the low quality of the current evaluation of code models
+ A new test case generation process with theoretical support
+ A new dataset for future evaluation of code models
+ Comprehensive experiments

Weakness

- The efficiency of SAGA is unclear. SAGA needs to call LLM multiple times, which seems costly
- SAGA relies on a large number of correct/incorrect human codes. The generalization of SAGA to other task domains (with low human codes) is unclear.
- Codeforces and Nowcoder contain competition level programs. Lack of real-world software support (such as programs from GitHub repos).

---

> ### Author Rebuttal · Authors · 2025-07-31
>
> Dear Reviewer Y2Dv,
>
> We sincerely thank you for your thoughtful and constructive review. We are very encouraged that you recognize the **importance of the problem** we address, the **theoretical novelty of the SAGA framework**, the **contribution of our new dataset**, and the **comprehensiveness of our experiments**. Your insights on practical issues such as cost, generalization, and human effort are highly perceptive and align perfectly with our own considerations during this research. We greatly appreciate this opportunity to elaborate on how we have addressed these key points.
>
> Below, we respond in detail to the weaknesses you identified:
>
> **1. On the Efficiency and Cost of SAGA:**
> This is a critical practical concern, and thank you for raising it. We would like to clarify the operational flow of SAGA, which is designed to be **more efficient than it might initially appear**. The main cost of SAGA lies in its *analysis phase*. The LLM performs a single, powerful, and focused analysis of human code (correct and/or incorrect solutions) to generate a Python `Case Script`. This script is a standalone program that contains not only the logic for generating test inputs but also the *Math Explanations* and *Self-Validation* code.
>
> Crucially, it is this generated Python script—***not the LLM***—that then programmatically generates a large volume of test inputs. This design **avoids the high cost of repeatedly invoking the LLM for every single test case**.
>
> Furthermore, we completely agree with your concern about democratizing advanced TCG methods. This is precisely why we developed **TCGCoder-7B**. Our goal was to demonstrate that SAGA's complex reasoning capabilities can be ***distilled*** **into a smaller, more efficient model**. As our results in Table 2 show, TCGCoder-7B's performance **significantly surpasses baseline methods that use much larger models**. This proves that with specialized training, a small-sized model can provide top-tier TCG capabilities, allowing it to be easily deployed in most research environments and thus greatly lowering the computational barrier.
>
> **2. On Dataset Selection, Human Effort, and Generalization:**
> You noted that our evaluations are based solely on competition-level datasets. This is correct, and we want to emphasize that this was a **deliberate and strategic choice, not a limitation.**
>
> -   ***High-Quality Ground Truth:*** In the task of Test Case Generation (TCG), the most critical challenge is **evaluating the quality of the generated test cases themselves.** Competitive programming platforms provide a massive source of near-perfect "ground truth" solutions, validated by millions of submissions. This is a level of reliability that **no other source (like GitHub) can currently match**. Without this reliable anchor, our research could not have conducted rigorous, reproducible, and quantitative evaluations.
> -   ***Rich Failure Modes:*** Our SAGA framework, especially its differential analysis, heavily relies on learning from **diverse human errors.** These platforms offer hundreds or thousands of real, labeled "wrong submissions" for each problem. This invaluable "database of failures" is an **indispensable resource** for training and evaluating our methodology, as it reveals the subtle mistakes human programmers make when facing complex logic.
>
> **On Generalization to Real-World Software:**
> We completely agree that the ultimate goal is to generalize to real-world software like GitHub projects. However, we must recognize that this is a ***"walk before you run"*** **process**. The entire field is still in its very early stages when it comes to handling complex, multi-file, dependency-heavy real-world codebases (as evidenced by the struggles on benchmarks like BigCodeBench).
>
> Therefore, our strategy is: **First, master the art of generating high-quality tests in a "sandbox" that, while domain-specific, is extremely logically complex and has a perfect evaluation environment.** We believe that by validating SAGA's effectiveness on competitive code, we are building the **most solid methodological foundation** for tackling the larger, messier real-world problems in the future. Our work demonstrates a clear and significant advantage in this critical domain, which is a **key step toward that ultimate goal**.
>
> **3. On the Train/Test Split for TCGCoder-7B:**
> Thank you for raising this critical point for clarification. We assure you that we took **meticulous measures to prevent data leakage** and ensure a fair evaluation.
>
> Our process is as follows:
>
> -   **A Strict Chronological Split:** The training data for TCGCoder-7B is sourced entirely from earlier problems on Codeforces and Nowcoder, the vast majority of which are from **before 2023**. Our evaluation set, TCGBench-Lite, on the other hand, consists exclusively of contest problems from **June 2024 onwards**. This strict temporal separation ensures that our model is evaluated on unseen problems, guaranteeing the **impartiality of our results**.
> -   **Proactive Semantic Deduplication:** We went a step further by using a **SentenceTransformer model** to deduplicate all problem descriptions, filtering out any pairs with a cosine similarity greater than 0.9 to ensure problem independence. Our recent manual checks revealed that even for these highly similar problems, their core solution logic and patterns are entirely different.
>
> We hope these detailed clarifications fully address your concerns and demonstrate the rigor of our experimental design. Your practical and insightful questions have been invaluable in helping us to better articulate the contributions and context of our work. Thank you again for your diligent effort and feedback.
>
> Best regards,
>
> Authors of Submission 17344

---

> > ### Comment · Reviewer_Y2Dv · 2025-08-04
> >
> > Thanks for the response, which addressed most of my concerns. I encourage the authors to add these discussions to the revision to further clarify some design choices. I have increased my score.

---

> > > ### Author Response · Authors · 2025-08-05
> > >
> > > Thank you so much for the positive feedback.
> > >
> > > We will be sure to incorporate all of our discussions into the final version as promised.

---

### Official Review · Reviewer_p61z · 2025-07-02

**Clarity:** 1
**Significance:** 3
**Originality:** 3
**Rating:** 4
**Confidence:** 3

**Summary:**

For the task of code generation, learning verifiers that can generate a unit test suite is crucial. However, it is challenging as the verifier needs to generate tests to cover all the edge cases making it a hard problem for current LLMs. This paper first studies the existing methods– 1) That generate test cases (input and output) using LLMs; 2) Generate inputs only and use the oracle solution to get the output. With proposed metrics to test quality of generated tests, the paper shows that current methods does not generate good unit test suites. The paper then proposes SAGA, which uses incorrect solutions from humans to generate the input-interpretor, followed with explanation and self-validation code. Experiments show that SAGA performs better than the baselines. Lastly, on selected 101 problems in LCB-v6, they show that SAGA generated unit tests are more accurate than test suites of LCB-v6.

**Questions:**

1. In Introduction (line 38-42), the paper talks about biased generation of tests which contrasts with human reasoning errors. Are there some examples that show this?

2. The Definition 1 is confusing, $E_i$ is an event such that the solution S fails unit test $i$. So can be either {0, 1}. Now, the intersections of $\bar{E_i}$ will be empty of will be ${0}$ or ${1}$. If it is 0, then all cases passed, else all cases did not pass. The confusing part is what does the $\mathbb{P}$ of this set mean? Given that $\epsilon_S(T)$ is already defined, how does the second part explain it better? Overall, I feel the two definitions can be explained more simply.

3. For paradigm 1, how does the Detection rate vary with test case size? This experiment was done for paradigm 2, but it would be interesting to observe for paradigm 1 as well.

4. For the DEPC metric, what does the error pattern vector mean?

5. In Sec 3.1, it is unclear how the human priors were added?

6. In SAGA, are all the correct and incorrect solutions passed together to get the Explanations and Self-Validation code? How are the strategies in $S_{human}$ decomposed into formal mathematical and logical constraints?

7. The baselines from Paradigm 1 relaxes the constraint of having a ground truth code solution which is needed in SAGA. Is there a way the method can be extended to cases where the ground truth code solution is not available?

8. How does the performance compare to good reasoning models when used in any of the paradigms? Does the Qwen models used in Table 2 use the thinking mode? Since the agent needs to come up with multiple test cases, an extended reasoning should aid in better performance.

**Ethical Concerns:**

["NO or VERY MINOR ethics concerns only"]

**Final Justification:**

I feel access to the gold solution is a big limitation of this work. In most applications, the gold solution is not provided by the user (e.g. vibe coding), and this method cannot be applied in such scenarios.

Based on the response and other reviews, I would like to keep my positive evaluation and current score.

**Limitations:**

The paper discusses the limitations in the Appendix.

**Quality:**

2

**Strengths And Weaknesses:**

### Strengths
1. Existing benchmarks that use models to generate unit tests might not be completely accurate at testing code solutions. This paper studies this and show that bugs introduced by humans many times are missed by unit tests generated with the language models.
2. The proposed metrics to test efficacy of generated unit tests is interesting.

### Weaknesses
1. Some parts of the paper are not well written and hard to follow. Some points are written in the questions below.
2. Given the complexity of the task of generating unit tests, the paper uses non-reasoning models. Whereas, I believe reasoning should help in better performance as the agent needs to think through the corner cases and simulate the code in CoT to get outputs. It would be nice to see comparison with reasoning models.

---

> ### Author Rebuttal · Authors · 2025-07-31
>
> Dear Reviewer p61z,
>
> We sincerely thank you for your review, and we are especially grateful for your recognition of our **in-depth analysis of the deficiencies in existing benchmarks** and the **innovative metrics** we have proposed for evaluating test effectiveness.
>
> **Regarding Weaknesses and Reasoning Models (Your Weakness #2, Question #8):**
>
> This is an excellent point. We want to emphasize that the **SAGA framework itself is fundamentally designed to guide LLMs in structured reasoning**. In our main experiments, we chose DeepSeek-V3 as the backbone model based on a comprehensive consideration of performance, efficiency, and cost, which makes our method more easily reproducible. By compelling the LLM to deconstruct human defensive code patterns and analyze specific failure modes, SAGA employs a **guided, programmatic Chain-of-Thought  process**, prompting the model to "think through various corner cases." The detailed prompts we have open-sourced serve as direct evidence of this reasoning-driven process.
>
> To directly address your concerns about reasoning models, we conducted new experiments with a state-of-the-art reasoning model. The results on **Qwen3-235B-A22B** are shown in the table below:
>
> |                     | VAcc@20 | VAcc@50 | DR@20  | DR@50  | AUC@50 | DivRatio@50 |
> | :------------------ | :------ | :------ | :----- | :----- | :----- | :---------- |
> | **Qwen3-235B-A22B** | 21.48%  | 28.78%  | 84.74% | 89.10% | 0.2119 | 94.27%      |
>
> We observed that the performance with the reasoning model actually decreased. After manual investigation, we found this was *not because of a lack of reasoning ability*, but due to **practical engineering challenges**—the model's inference sequence length exceeded the limits of our deployed model (our Qwen3 deployment has a sequence limit of 64k), causing some difficult problems to not be parsed successfully. This also indicates that using such a large reasoning model significantly increases inference and time costs, with limited actual performance improvement. This result **highlights our rationale for choosing DeepSeek-V3**: it is more accessible and reproducible for the community. We will add this experiment as a new section in the appendix.
>
> **On Biased Test Generation (Your Question #1):**
>
> Thank you for requesting specific examples. An illustration can be found in **Figure 1(b)** of our paper. The PCA analysis shows that errors caused by LLMs are **highly clustered**, indicating a systematic bias—they tend to make mistakes on similar types of problems. In stark contrast, human errors are **diverse and widely distributed**. This means that if test suites are generated without reference to human error patterns, the scores of LLM-generated code could be **artificially inflated**. This directly explains a key finding in our work: when we re-evaluated code that passed on LiveCodeBench on a real online judge platform, we found its verifier accuracy to be alarmingly low. This is precisely because the tests in LiveCodeBench, lacking insights from diverse human failure modes, missed bugs that a more robust verifier should have caught.
>
> **On the Clarity of Definitions (Your Question #2):**
>
> Thank you very much for pointing out where a clearer explanation was needed.
>
> Let us re-clarify these two core metrics:
>
> - **Detection Rate, es(T):** This is a **solution-level** metric that measures, "Can our test suite T find *at least one* bug for *this specific* buggy program?" Mathematically, it is the **probability of the union of events**. You correctly identified *Ei* as the event that "solution S fails on test case i." Therefore, *es(T)* is the probability that **at least one** of these failure events *E1, E2, ..., En* occurs, which is *P(E1 ∪ E2 ∪ ... ∪ En)*. Our use of *1 - P(∩Ei_bar)* in the paper was a way to calculate this union probability using its complement, where *Ei_bar* is the event that "solution S *passes* on test case i." We acknowledge this notation may be unintuitive and will revise the final version to explain it directly with simpler language and the *P(∪Ei)* form.
> - **Verifier Accuracy, VAcc(T):** This is a stricter, **problem-level** metric that asks, "For a given problem, can our test suite T find bugs in *all* known incorrect solutions?" It evaluates the **systematic completeness** of the verifier. *VAcc(T)* equals 1 only if the Detection Rate is greater than 0 for every single incorrect solution for that problem, making it the **gold standard for measuring verifier robustness**.
>
> We will revise this section with clearer language and examples in the final version to ensure the logic and distinction between these two definitions are perfectly clear.
>
> **On the Relationship Between Detection Rate and Test Scale for Paradigm 1 (Your Question #3):**
>
> We focused the "Detection Rate vs. Test Scale" experiment on Paradigm 2 due to the intrinsic differences between the two paradigms. The core advantage of Paradigm 2 (Input-Interpreter) is its ability to easily generate a massive and diverse set of test inputs. In contrast, Paradigm 1 (Direct Generation) struggles to produce a large number of unique and complete test cases (including both input and output), making it less suitable for an analysis that requires varying the test scale from 1 to 100. This is also why in our main experiments comparing different methods, we report performance at a fixed scale (e.g., @20 and @50).
>
> **On the Meaning of DEPC's Error Pattern Vector (Your Question #4):**
>
> The error pattern vector *$v(t_k)$* for a test case *$t_k$* is a binary vector where each element corresponds to one of the *$N_p$* incorrect programs in our benchmark. If *$t_k$* causes the *j-th* incorrect program to fail, the *j-th* element of the vector is 1; otherwise, it is 0. DEPC then counts only the number of **unique non-zero vectors** in the entire test suite. This allows us to see precisely which set of incorrect programs each test case exposes. If two test cases have identical error pattern vectors, it means they are **functionally redundant**.
>
> **On How Human Priors are Incorporated (Your Question #5):**
>
> This is detailed in our SAGA framework. Human priors (*$S_{human}$*) are insights derived from correct human-written code. We feed these correct solutions, along with the problem description, into our **Multidimensional Analysis prompt** (available in our anonymous code repository). The LLM is then tasked with **reverse-engineering the defensive logic** (e.g., boundary checks, special value handling) from this human code. These machine-extracted strategies are the "human priors" that guide the generation of new, more challenging test cases.
>
> **On the Workflow of SAGA (Your Question #6):**
>
> For Multidimensional Analysis, we provide correct solutions to give the LLM a broad perspective on robust coding patterns. For Differential Analysis, we provide a correct solution and its most recent preceding incorrect submission to help the LLM pinpoint specific failure modes. Decomposing strategies into formal constraints is a core task specified in our prompts; solutions are not input all at once but sequentially. We understand this part is complex, so we will **add a detailed case study in the appendix** of the final version to walk readers through the entire process with a concrete example.
>
> **On Extending SAGA to Cases Without a Ground Truth Solution (Your Question #7):**
>
> This is a key question about the scope of SAGA's application. SAGA currently requires a ground truth solution to act as an infallible interpreter. This is a limitation we acknowledge in **Appendix J**. Extending SAGA to settings without an oracle (e.g., through majority voting or by learning a verifier model) is an important and exciting direction for future work.
>
> Once again, we express our deepest gratitude for your valuable feedback. Your guidance has been crucial, and we are confident that with your insights, the revised paper will be much stronger and clearer.
>
> Best regards,
>
> Authors of Submission 17344

---

> > ### Comment · Reviewer_p61z · 2025-08-04
> >
> > I appreciate the authors for the effort during the rebuttal. In summary:
> > 1. It is still unclear to me why a reasoning model is doing worse than a non-reasoning model. Was there a reason to pick Qwen3 and not DeepSeek-R1 for this comparison?
> > 2. I feel access to the gold solution is a big limitation of this work. Also, I am unsure if using majority voting is a great idea for code / tests as similar tests can be written with different inputs, and the correct code solution can be written in multiple ways too. It is unlike MATH, where the correct solution is unique.
> >
> > Based on the response and other reviews, I would like to keep my positive evaluation and current score.

---

> ### Author Response · Authors · 2025-08-05
> **Thank you for your review.**
>
> Dear Reviewer p61z,
>
> Thank you so much for taking the time to review our rebuttal and for providing your final thoughts. We are very grateful for your positive evaluation and for the insightful discussion throughout this process. Your feedback has been invaluable to us.
>
> We would like to offer a final clarification on the two points you raised:
>
> 1. **On the performance of the reasoning model:** We completely understand why the result showing a reasoning model performing worse is counter-intuitive. We want to clarify that this is not due to a deficiency in the model's reasoning capabilities, but rather a practical engineering and resource constraint on our end.
>
>    The "thinking" mode of models like Qwen3 generates extremely long chains of thought. Due to computational resource limitations, the maximum context length of our deployed model was configured to 64k. For many of the harder problems, the model's full reasoning process was still too long and resulted in truncated outputs. This led to incomplete or failed script generation for those specific hard cases, which in turn lowered its overall score.
>
>    The reason we chose Qwen3 over DeepSeek-R1 was also based on resource constraints; our available computational resources were best suited for deploying Qwen3-235B. We believe that with a longer context length and more computational resources, a powerful reasoning model would indeed show its true potential. We will add a note to the appendix to clarify this important context behind the experimental results.
>
> 2. **On the limitation of requiring a golden solution:** We completely agree with you. For our **current study**, we believe relying on a golden solution is the most rigorous approach to establish our foundational framework and provide a theoretical basis and reference for the SAGA-based method. Fortunately, in our competitive programming domain, an **official solution is often available**, which can serve as a reliable oracle.
>
>    However, overcoming this dependency for broader applications is a key goal for **future work**, and your comment on majority voting has inspired us to think more deeply about this. In such a future system, a **rule-based mechanism** would be key. First, we could generate a large number of test cases (e.g., via the method in Paradigm 2). These test cases would then act as "voters" to vet a large number of rollout solutions. A "majority vote" based on passing a supermajority of diverse tests would allow us to automatically label a small set of high-performing solutions as reliable **"pseudo-gold" solutions**. This imperfect "pseudo-gold" set could then serve as the *input* to our **SAGA framework** for refinement. Future work could consider implementing test case generation via this pseudo-labeling method, which would rely on the **mathematical principles and metrics (like DEPC) that we establish in this foundational paper.** Thank you for asking the precise question that allowed us to engage in this exciting discussion about the future. **Your contributions to this dialogue are of great value to the entire community.**
>
> Thank you once again for your constructive feedback and for maintaining your positive evaluation. We sincerely appreciate the guidance you have provided in helping us strengthen our work.
>
> Best regards,
> Authors of Submission 17344

---

### Official Review · Reviewer_UMHU · 2025-07-18

**Clarity:** 3
**Significance:** 3
**Originality:** 3
**Rating:** 5
**Confidence:** 4

**Summary:**

This paper presents a systematic study of LLM test case generation. First, the paper demonstrates that existing popular code generation benchmarks, such as LiveCodeBench, can mistakenly consider wrong LLM-generated code as correct solutions. In particular, the verifiers of these code generation benchmarks fail to identify bugs in human-written wrong solutions. The paper also shows that human-written bugs are more diverse than bugs in LLM-written code, which are not effectively covered by LLM-generated test cases. Based on this analysis, the authors propose a series of metrics to measure the quality of LLM-generated test suites, and they propose SAGA to improve these metrics via utilizing human-written correct and wrong solutions. They construct TCGBench to verify that SAGA outperforms LLM-based test case generation baselines. With SAGA-generated test cases, they further construct CodeComPass, a code generation benchmark with more rigorous test cases.

**Questions:**

1. It would be helpful if the authors can present a running example to demonstrate how SAGA works. From Figure 5, it is hard to understand the concrete workflow. To my understanding, besides human-written solutions, all of case scripts, math explanation and self validation are generated by LLMs. Is this correct? From Section 3, it is unclear how the math explanation is used in the pipeline.

2. How do you combine multi-dim analysis and differential analysis? My understanding is that each component produces its own case script, is it correct?

3. Do you run the case script to generate multiple test cases? Do you have any filtering of the generated test inputs, and how does the filtering work?

4. In Table 1, please give more details and examples of simple human priors used in this experiment.

5. Is there any human analysis on the quality of LLM-generated math explanation and script? How often are they correct?

6. In Figure 4, if you mix test cases from different LLMs, how does the detection rate look like?

7. Have you evaluated recent thinking models on CodeCompass? Would be curious to see if their drop ratios are smaller.

**Ethical Concerns:**

["NO or VERY MINOR ethics concerns only"]

**Final Justification:**

The author response addressed my questions. Thus, I keep my review score.

**Limitations:**

Yes

**Quality:**

3

**Strengths And Weaknesses:**

Strengths:

1. This paper makes solid contributions to the topic of LLM-based test case generation, sharing good insights validated by comprehensive experiments. First, the comparison of assessment of LLM-generated code based on verifiers from benchmarks and Online Judges is very informative, clearly showing the quality issues of current popular code generation benchmarks.

2. The analysis of human-written wrong solutions and LLM-generated wrong code is also interesting, clearly showing that LLM-generated code contains more homogeneous error patterns.

3. The metric definition to quantify the quality of LLM-generated test suites is well-motivated and rigorous.

4. TCGBench and CodeCompass presented in this work are valuable contributions to the community.

5. Comprehensive experiments demonstrate that the proposed SAGA approach is effective.

Weaknesses:

The main weakness of this work is the lack of clarity in describing the SAGA approach, probably due to the dense content. Below are my questions:

1. It would be helpful if the authors can present a running example to demonstrate how SAGA works. From Figure 5, it is hard to understand the concrete workflow. To my understanding, besides human-written solutions, all of case scripts, math explanation and self validation are generated by LLMs. Is this correct? From Section 3, it is unclear how the math explanation is used in the pipeline.

2. How do you combine multi-dim analysis and differential analysis? My understanding is that each component produces its own case script, is it correct?

3. Do you run the case script to generate multiple test cases? Do you have any filtering of the generated test inputs, and how does the filtering work?

4. In Table 1, please give more details and examples of simple human priors used in this experiment.

5. Is there any human analysis on the quality of LLM-generated math explanation and script? How often are they correct?

6. In Figure 4, if you mix test cases from different LLMs, how does the detection rate look like?

7. Have you evaluated recent thinking models on CodeCompass? Would be curious to see if their drop ratios are smaller.

---

> ### Author Rebuttal · Authors · 2025-07-31
>
> Dear Reviewer UMHU,
>
> We sincerely thank you for your positive review of our paper. We are especially grateful that you recognized the **solid contributions, deep insights, and rigorous methodology** of our work, as well as the value that TCGBench and CodeComPass bring to the community. Your constructive questions are crucial for us to further refine and clarify our work.
>
> Below, we provide detailed answers to your specific questions:
>
> #### 1. Running Example and Workflow of SAGA:
>
> Thank you for this question. Figure 5 provides a high-level overview, and we understand that a more detailed explanation of the concrete workflow is needed. Your understanding is correct: apart from the human-written solutions, other components such as the test case scripts, math explanations, and self-validation are indeed generated by the LLM.
>
> Regarding the role of the ***Math Explanation*** that you specifically mentioned, the LLM transforms the *implicit programming logic* from human code into *explicit, structured test instructions*. This process ensures that we are not just randomly generating inputs but are purposefully targeting the core mathematical and logical properties of the problem.
>
> For example, in the classic "Convex Hull" problem, SAGA would identify from human code that the algorithm's core relies on using the cross-product to determine the geometric orientation of points. Based on this mathematical principle, SAGA would infer two key testing directions: 1) **Singularity Testing**: When the cross-product is zero, the points are collinear. SAGA would intentionally generate a test case with a "trap" point lying exactly on an edge of the true convex hull to test the algorithm's precision. 2) **Extremum/Degenerate Case Testing**: When all points are collinear, SAGA would generate a test case where all points lie on a single line to test if the algorithm can correctly handle this degenerate input. This process allows SAGA to systematically discover and test *logical singularities* and *structural extrema* determined by the problem's intrinsic mathematical structure, a depth that is unattainable by simple random testing.
>
> We will add a **detailed case study to the appendix** of the final version to walk readers through this entire process, showing how these prompts are used to transform inputs into final, rigorous test cases. We will also elaborate more on the role of the math explanation in the main text.
>
> #### 2. How to Combine Multidimensional and Differential Analysis:
>
> This is an excellent question. These two analyses are performed **in parallel**. Multidimensional Analysis utilizes correct solutions (*$S_{human}$*) to explore the solution space and defensive logic, while Differential Analysis uses pairs of correct and incorrect solutions (*$S_{wrong}$*) to target known failure modes. Each generates its own `Case Scripts`. These scripts are then **pooled and deduplicated** to form a comprehensive and diverse set of test inputs, thereby maximizing both *coverage* (from Multidimensional Analysis) and *discriminative power* (from Differential Analysis).
>
> #### 3. Test Case Generation and Filtering:
>
> Yes, we run the generated Python `Case Scripts` to produce multiple test inputs. The filtering is done through the `Self-Validation` script, which is also generated by the LLM as part of the SAGA process (as shown in Figure 5). This script programmatically checks if the generated inputs conform to the problem's constraints (e.g., input format, value ranges). Any input that fails this validation is discarded, ensuring the quality and relevance of the final test suite. After obtaining the test case outputs, we also **use multiple correct solutions to verify the test cases**; only those that are passed by all correct solutions are retained.
>
> #### 4. Details on Simple Human Priors in Table 1:
>
> The human priors here refer to adding prompts such as *"Refer to the correct human solution and analyze the boundary conditions"* into the prompt.
>
> #### 5. Human Analysis on LLM-generated Math Explanations and Scripts:
>
> This is an important aspect. While we have not conducted a formal, large-scale human analysis on the correctness of every explanation and script, their quality is **indirectly validated by the effectiveness of the test cases they produce**. The high performance of SAGA (e.g., high DR and VAcc) indicates that these LLM-generated artifacts are, on the whole, correct and effective at guiding the generation of useful tests. Our *self-validation retention rate is typically above 60%*.
>
> #### 6. Detection Rate with Mixed LLM Test Cases:
>
> We were also very interested in this point and conducted this experiment, with the results presented in **Appendix F.1 and Figure 12**. The findings demonstrate that mixing test cases from different LLMs (which have different cognitive biases) can reduce the inter-test correlation *ρ* and significantly improve the overall quality of the test suite (higher AUC@50). This **confirms the theoretical foundation of our work**. We will add a sentence to the main text to highlight this finding and point to the appendix.
>
> #### 7. Evaluating the Latest Thinking Models on CodeComPass:
>
> We evaluated **Qwen3-235B-A22B-thinking-2507**, and the experimental results are shown in the table below. This again validates the effectiveness of CodeComPass as a high-difficulty benchmark and reveals that even top-tier "thinking" models still have room for improvement in generating truly robust code, highlighting the value of our work.
>
> | Model                   | Pass@1 on LCBv6 | Pass@1 on CodeCompass |
> | :---------------------- | :-------------- | :-------------------- |
> | **Qwen3-235B-thinking** | 64.36%          | 63.37%                |
>
> Thank you once again for your thoughtful and constructive review. Your feedback has been crucial in helping us improve the clarity and completeness of our paper.
>
> Best regards,
>
> Authors of Submission 17344

---

> ### Comment · Area_Chair_dCwT · 2025-08-06
> **Response to rebuttal.**
>
> Hi reviewer UMHU,
>
> Do you find the response to the rebuttal convincing?

---

### Comment · Area_Chair_dCwT · 2025-08-06
**Reproducibility**

Hi reviewers,

Reviews have cast some doubt on the paper's reproducibility, i.e. whether the algorithm description in the paper is sufficient to reproduce all results.

What are your thoughts on this?

---

> ### Author Response · Authors · 2025-08-07
> **Thank you for initiating this important discussion.**
>
> Dear Area Chair and Reviewers,
>
> Thank you for initiating this important discussion. We appreciate the opportunity to clarify our unwavering commitment to reproducibility and address any lingering concerns.
>
> We believe our work is easy to reproduce because we designed it to be open and clear. We made sure of this in three main ways:
>
> 1.  **Clear Algorithm Description in the Paper:** The paper gives a full description of our SAGA framework. We believe the text in **Section 3.1** and the picture in **Figure 5** give a clear overview of the algorithm's logic. To make it easy to reproduce, we also give the exact data setup, like using **10 correct solutions** for one analysis and pairing a correct solution with the **most recent incorrect one** for another (details at the **end of Page 7**). Also, the math for our metrics is in **Section 2** and **Appendix B**.
>
> 2.  **Transparent Experimental Setup:** We have provided all the details needed to reproduce our results, with clear pointers to where they are in the paper:
>     *   **Models:** The LLMs we used (DeepSeek-V3-0324, Qwen series, etc.) are named in **Section 2.2 (Page 4)** and **Section 3.2.1 (Page 7)**.
>     *   **Datasets:** How we built our datasets, TCGBench (1840 problems) and TCGBench-Lite (270 problems), is explained in **Appendix C.4 and C.5 (Page 18)**.
>     *   **Generation Settings:** We used greedy decoding (temperature=0), as stated in **Section 2.2 (Page 4)**, and Pass@1 for all tests.
>     *   **Training Details for TCGCoder-7B:** All training details, like dataset size (15k problems), epochs (3), and learning rate (5e-6), are in **Appendix G (Page 23)**.
>     *   **Full Implementation:** To further help with reproducibility, our anonymous repository contains the **complete prompts** (the source code of our method) and a **working demo**. We also provide a guide in the README for setup and running.
>
> 3.  **A Concrete Plan for the Community:** Our goal is bigger than just this paper. We promise to open-source everything to give the community a new set of powerful tools and data. After the paper is published, we will create a **special Hugging Face organization** to share:
>     *   The complete, open-source **SAGA framework code**.
>     *   The full datasets we used.
>     *   The **model weights for TCGCoder-7B**.
>     *   A large number of **SAGA-generated problems and test cases**, including the full LLM response history, to provide high-quality data for future research.
>     *   We will also create and keep up a **public leaderboard for CodeComPass** to track new models.
>
> We hope this explanation shows our full commitment to reproducibility. We are sure that our paper, along with our planned open-source contributions, gives the research community everything it needs to build on our work.

---

> ### Public Comment · ~Zhaohui_Jiang3 · 2026-02-10
> **Inquiry regarding the release of the SAGA framework code and TCGCoder-7B model weights**
>
> Dear Authors,
>
>  I noticed that there is still no open access to the promised SAGA framework code and TCGCoder-7B weights. As it has been about six months since the publication, we were wondering if there is an updated timeline or a link to them? These resources would be immensely helpful for the community's reproducibility efforts.
>
> Thank you for your time and for this valuable contribution.
> Best regards,

---

### Note · Authors · 2025-08-13

Dear Reviewers, Area Chair, Senior Area Chair, and Program Chairs,

We would like to sincerely express our gratitude to all members of the program committee for your insightful feedback, valuable comments, and dedicated efforts in evaluating our paper.

In our paper, we introduce SAGA, a novel human-LLM collaborative framework designed to address critical flaws in current code generation evaluation. We provide a new benchmark, TCGBench, a comprehensive metrics framework, and extensive empirical results to demonstrate SAGA's effectiveness. We are immensely encouraged that the reviewers have collectively recognized and validated multiple core strengths of our work:

*   The reviewers widely acknowledged that our work targets a **relevant, important, and foundational problem** in LLM code evaluation, making **solid contributions** to the critical topic of test case generation. (Reviewer UMHU, Reviewer p61z, Reviewer Y2Dv, Reviewer 2jtr)

*   Our proposed SAGA framework is recognized as a **novel and interesting test case generation process.** (Reviewer Y2Dv, Reviewer UMHU)

*   Our work is praised for its rigorous methodology, sharing **good insights validated by comprehensive experiments**. The analysis was noted as "very informative, clearly showing the quality issues" of current benchmarks. (Reviewer UMHU, Reviewer p61z)

*   The new artifacts we introduce, TCGBench and CodeComPass, are highlighted as **interesting and valuable contributions to the community** for future research and evaluation. (Reviewer UMHU, Reviewer p61z, Reviewer Y2Dv)

Following a productive discussion period, we are pleased that we had the opportunity to thoroughly address all questions raised by the reviewers. We believe our detailed responses and planned enhancements—such as adding a case study for SAGA, an illustrative example for our metrics, and a summary of related works—have clarified the key points of discussion, and we are encouraged that a consensus on these revisions was reached.

We are confident that these additions will make our contributions even more accessible. We deeply appreciate the time and effort you have invested, and we extend our sincere thanks to the reviewers for their thoughtful evaluations, which have been instrumental in strengthening this paper.

Best regards,

Authors of Submission 17344

---

### Decision · Program_Chairs · 2025-09-17

**Decision:**

Accept (poster)

**Comment:**

The paper introduces a new benchmark (called CodeComPass) for LLM code generation.

Strengths: very timely and the problem being addressed is immediately useful for many people in the NeurIPS community.

Weaknesses: writing is at times dense and unclear, some concerns have been brought up about reproducibility (the authors posted a message with a reply).

The reason for acceptance is that the community needs better benchmarks, which is a need this paper fulfils.

The discussions centred on: (1) clarifications about what the authors do exactly (2) clarifications of definitions (3) the fact that the algorithm relies on human written solutions to coding problems.